

# Ensemble modeling of stochastic unsteady open-channel flow in terms of its time-space evolutionary probability distribution: numerical application

Alain Dib and M. Levent Kavvas

Department of Civil and Environmental Engineering, University of California, Davis, 95616, USA

*Correspondence to*: Alain Dib (aedib@ucdavis.edu)

**Abstract.** The characteristic form of the Saint-Venant equations was solved in a stochastic setting by using a newly proposed Fokker–Planck Equation (FPE) methodology. This methodology computes the ensemble behavior and variability

of a system by directly solving for its time-space evolutionary probability distribution. The new methodology was tested on a stochastic unsteady open-channel flow problem, with an uncertainty arising from the channel's roughness coefficient. The computed statistical descriptions of the flow variables were compared to the results obtained through Monte Carlo (MC) simulations in order to evaluate the performance of the FPE methodology. The comparisons showed that the proposed methodology can adequately predict the results of the considered stochastic flow problem, including the ensemble averages,

variances, and probability density functions in time and space. However, unlike the large number of simulations performed by the MC approach, only one simulation was required by the FPE methodology. Moreover, the total simulation period of the FPE methodology was significantly smaller than that of the MC approach. As such, the results obtained in this study indicate that the proposed FPE methodology is a powerful and time-efficient approach for predicting the ensemble average and variance behavior, in both space and time, for an open-channel flow process under an uncertain roughness coefficient.

## 1 Introduction

One of the most important types of unsteady open-channel flow problems is that of flood routing (Scharffenberg and Kavvas, 2011). This problem considers an initially uniform flow rate through an open channel, after which a flood wave enters upstream of the channel reach, and is translated downstream. The routing process, which predicts the spatial shape and temporal development of this flood wave as it traverses downstream (Viessman et al., 1977), involves solving for two

dependent flow variables as a function of time and space through the river reach: velocity and depth, or discharge and depth. Solving for these two unknowns by means of the hydraulic routing technique involves using two governing equations (Chow, 1959), the continuity equation and the momentum equation, which are jointly known as the Saint-Venant equations (Sturm, 2001).



While it may be possible to assume that the flow and channel parameters are deterministic so as to obtain a deterministic solution to the Saint-Venant equations, such parameters usually exhibit high uncertainty in the real world (Gates and AlZahrani, 1996; Ercan and Kavvas, 2012a). In fact, uncertainties in these parameters may originate from several factors, including a channel's physical conditions and geometric parameters, its upstream boundary and initial conditions, as well as any lateral inflows (Chow, 1959; Sturm, 2001; Liang and Kavvas, 2008; Ercan and Kavvas, 2012a). With such uncertainties, the parameters become spatially and/or temporally random, in which case the system can no longer be assumed deterministic. This necessitates the solution of the governing equations within a stochastic framework, from which a quantitative description of the ensemble behavior and variability of the process is obtained. In this manner, the two dependent flow variables are solved for their statistical properties, not for their deterministic values, at designated time-space positions.

Among the available approaches that can be used to solve the Saint-Venant equations within a stochastic framework, the Monte Carlo (MC) approach is one of the most well-known due to its abundant use in simulating differential equations with stochastic parameters (Freeze, 1975; Smith and Freeze, 1979; Bellin et al., 1992). It is also generally accepted as the most robust approach for uncertainty evaluation, as well as the benchmark for comparing other new methods (Scharffenberg and Kavvas, 2011). However, one of the main disadvantages of the MC approach is its computational expense, which results from the large number of simulations that it usually involves.

In order to circumvent having to solve the governing equations repeatedly for a large number of times, this study uses a new methodology that solves for the time-space evolutionary probability distribution of the system in only one simulation. From this probabilistic solution, one can then obtain the ensemble mean and ensemble variance of the process as they evolve in time and space. This new methodology is proposed, explained, and derived in the companion paper by Dib and Kavvas (2017). In short, the ensemble averaging technique developed in Kavvas (2003) is used to determine the deterministic equation for the evolutionary probability distribution of the governing stochastic differential equations of the flow process, thus providing their Lagrangian–Eulerian form of the Fokker–Planck equation (LEFPE). The LEFPE is then simplified to a classical Fokker–Planck Equation (FPE), which deterministically describes the temporal and spatial evolution of the probability density of the dependent variables of the flow process. Through an implicit discretization, the obtained FPE is solved numerically in order to compute the statistical properties of the dependent variables, thus describing the ensemble behavior and variability of the system being considered.

In addition to producing the complete ensemble model results in a computationally efficient manner by using one simulation, the proposed FPE methodology directly solves for, and is linear in, the probability density of the dependent variables. Note that while this methodology assumes a finite correlation time for the considered process, it does not make any linearization assumptions and it does not have limitations on the working range of the parameter space (Ercan and Kavvas, 2012a). Many hydrologic processes have been successfully simulated by using such a methodology, including unsaturated water flow (Kim et al., 2005a), root-water uptake (Kim et al., 2005b), solute transport (Liang and Kavvas, 2008),





snow accumulation and melt (Ohara et al., 2008), unconfined groundwater flow (Cayar and Kavvas, 2009a, b), as well as kinematic open-channel flow (Ercan and Kavvas, 2012a, b).

Therefore, in the wake of the preceding discussions, the first objective of this study is to apply the proposed FPE methodology derived in the companion paper (Dib and Kavvas, 2017) to a representative stochastic unsteady open-channel

flow problem in order to solve for the probability density of the state variables of the flow process, and to provide a quantitative description of the expected behavior and variability of the system in one single simulation. The second objective is to evaluate the performance of the proposed methodology and to validate its results by comparing the statistical properties of the flow variables computed by the FPE methodology against those calculated by the MC approach.

## 2 Saint-Venant equations: characteristic form and ensemble-averaged form

The Saint-Venant equations solved in this study are written for the unsteady open-channel flow of an incompressible fluid in a rectangular, prismatic channel with no lateral inflow. The method of characteristics is used to transform these equations from two partial differential equations into a system of four ordinary differential equations (ODEs). These four ODEs include two characteristic equations describing the two characteristic paths, and two compatibility equations which must be satisfied along their corresponding characteristic path. The characteristic form of the Saint-Venant equations is shown below

(Sturm, 2001):

*Positive characteristic curve ($C_1$)*
$$\frac{dx_1}{dt} = V + c \tag{1}$$

*Flow process condition to be satisfied along $C_1$*
$$\left(\frac{d(V + 2c)}{dt}\right)_1 = g\left(S_{0,1} - S_{f,1}\right) \tag{2}$$

*Negative characteristic curve ($C_2$)*
$$\frac{dx_2}{dt} = V - c \tag{3}$$

*Flow process condition to be satisfied along $C_2$*
$$\left(\frac{d(V - 2c)}{dt}\right)_2 = g\left(S_{0,2} - S_{f,2}\right) \tag{4}$$

where $V$ is the average flow velocity, $c$ is the wave celerity which is equal to $\sqrt{gy}$ for a rectangular channel where $y$ is the flow depth, $x$ is the position, $t$ is the time, $S_0$ is the slope of the channel bottom, $S_f$ is the friction slope, and g is the acceleration of gravity. Note that $S_{0,i}$ denotes $S_0(x_i,t)$; the same applies to $S_f$. The positive and negative characteristic curves are defined as $C_1$ and $C_2$, respectively, and the variables or derivatives corresponding to $C_1$ and $C_2$ are denoted by the

subscripts 1 and 2, respectively.

The LEFPE, corresponding to the Saint-Venant equations, that can solve for the multivariate probability density function (PDF) of the hydrologic state variables of an unsteady open-channel flow problem was obtained through the




following steps: (i) using the two substitutions $V + 2c = \alpha$ and $V - 2c = \beta$ on Eqs. (1) to (4), (ii) applying the technique in Kavvas (2003) while assuming the uncertainty arising from the Manning's roughness coefficient, and (iii) performing several simplifying assumptions. The result is shown in Eq. (5), while its detailed derivation can be found in the companion paper by Dib and Kavvas (2017).

$$
\begin{aligned}
\frac{\partial P(x_1, x_2, \alpha, \beta, t)}{\partial t} =& \\
&- \frac{\partial}{\partial x_1}\left\{ P\left[\frac{3}{4}\langle\alpha(x_1,t)\rangle + \frac{1}{4}\langle\beta(x_1,t)\rangle\right]\right\} \\
&- \frac{\partial}{\partial x_2}\left\{ P\left[\frac{1}{4}\langle\alpha(x_2,t)\rangle + \frac{3}{4}\langle\beta(x_2,t)\rangle\right]\right\} \\
&- \frac{\partial}{\partial \alpha}\left\{ P\left[g\,S_0 - \frac{g}{4k^2}\left(\frac{2}{b}\right)^{4/3}\left\langle n^2(x_1,t)\cdot[\alpha(x_1,t)+\beta(x_1,t)]^2\cdot\left\{\frac{8gb}{[\alpha(x_1,t)-\beta(x_1,t)]^2}+1\right\}^{4/3}\right\rangle\right]\right\} \\
&- \frac{\partial}{\partial \beta}\left\{ P\left[g\,S_0 - \frac{g}{4k^2}\left(\frac{2}{b}\right)^{4/3}\left\langle n^2(x_2,t)\cdot[\alpha(x_2,t)+\beta(x_2,t)]^2\cdot\left\{\frac{8gb}{[\alpha(x_2,t)-\beta(x_2,t)]^2}+1\right\}^{4/3}\right\rangle\right]\right\} \\
&+ \frac{\partial^2}{\partial x_1^2}\left\{ P\left[\left(\frac{9}{16}\right)\mathrm{Var}[\alpha(x_1,t)] + \left(\frac{1}{16}\right)\mathrm{Var}[\beta(x_1,t)] + \left(\frac{3}{8}\right)\mathrm{Cov}[\alpha(x_1,t),\beta(x_1,t)]\right]\right\} \\
&+ \frac{\partial^2}{\partial x_2^2}\left\{ P\left[\left(\frac{1}{16}\right)\mathrm{Var}[\alpha(x_2,t)] + \left(\frac{9}{16}\right)\mathrm{Var}[\beta(x_2,t)] + \left(\frac{3}{8}\right)\mathrm{Cov}[\alpha(x_2,t),\beta(x_2,t)]\right]\right\} \\
&+ \frac{\partial^2}{\partial \alpha^2}\left\{ P\left[\frac{g^2}{16k^4}\left(\frac{2}{b}\right)^{8/3}\mathrm{Var}\left[n^2(x_1,t)\cdot[\alpha(x_1,t)+\beta(x_1,t)]^2\cdot\left\{\frac{8gb}{[\alpha(x_1,t)-\beta(x_1,t)]^2}+1\right\}^{4/3}\right]\right]\right\} \\
&+ \frac{\partial^2}{\partial \beta^2}\left\{ P\left[\frac{g^2}{16k^4}\left(\frac{2}{b}\right)^{8/3}\mathrm{Var}\left[n^2(x_2,t)\cdot[\alpha(x_2,t)+\beta(x_2,t)]^2\cdot\left\{\frac{8gb}{[\alpha(x_2,t)-\beta(x_2,t)]^2}+1\right\}^{4/3}\right]\right]\right\}
\end{aligned}
\tag{5}
$$

5  In Eq. (5), $b$ denotes the width of the channel, $n$ denotes Manning's roughness coefficient, and k denotes the conversion factor between SI and US units for Manning's formula. The remaining variables are as defined previously. Note that in Eq. (5), and other later equations, $P(x_1, x_2, \alpha, \beta, t)$ is sometimes substituted by $P$ for simplicity. Moreover, note that Eq. (5) resembles an advection-diffusion equation, in which the first four bracketed terms multiplied by $P$ on their left-hand-sides resemble the advection coefficients, and the last four bracketed terms resemble the diffusion coefficients. Hence, after

10  denoting the advection coefficients by $F$ and the diffusion coefficients by $D$, Eq. (5) was simplified into Eq. (6) below, which is the final analytical form of the proposed FPE methodology for the stochastic solution of the Saint-Venant equations in one simulation.

$$
\begin{aligned}
\frac{\partial P(x_1, x_2, \alpha, \beta, t)}{\partial t} =& -\frac{\partial}{\partial x_1}F_1 P - \frac{\partial}{\partial x_2}F_2 P - \frac{\partial}{\partial \alpha}F_\alpha P - \frac{\partial}{\partial \beta}F_\beta P \\
&+ \frac{\partial^2}{\partial x_1^2}D_1 P + \frac{\partial^2}{\partial x_2^2}D_2 P + \frac{\partial^2}{\partial \alpha^2}D_\alpha P + \frac{\partial^2}{\partial \beta^2}D_\beta P
\end{aligned}
\tag{6}
$$



An appropriate numerical scheme to solve Eq. (6) was derived following Chang and Cooper (1970), as discussed in detail in the companion paper. This scheme involves implicitly discretizing the equation, while noting that the dependent variables are to be solved at the intersection of the characteristic curves $C_1$ and $C_2$, which renders $x_1 = x_2 = x$. As such, the discretized version of Eq. (6) was derived in the companion paper to be as follows:

$$
\begin{aligned}
P_{h,k,l}^{n} = {} & \left\{
\begin{aligned}
& 1 + \frac{\Delta t}{\Delta x}\delta_{1;\,h}^{n+1}F_{1;\,h+\frac{1}{2}}^{n} + \frac{\Delta t}{(\Delta x)^2}D_{1;\,h+\frac{1}{2}}^{n} - \frac{\Delta t}{\Delta x}\left(1 - \delta_{1;\,h-1}^{n+1}\right)F_{1;\,h-\frac{1}{2}}^{n} + \frac{\Delta t}{(\Delta x)^2}D_{1;\,h-\frac{1}{2}}^{n} \\
& + \frac{\Delta t}{\Delta x}\delta_{2;\,h}^{n+1}F_{2;\,h+\frac{1}{2}}^{n} + \frac{\Delta t}{(\Delta x)^2}D_{2;\,h+\frac{1}{2}}^{n} - \frac{\Delta t}{\Delta x}\left(1 - \delta_{2;\,h-1}^{n+1}\right)F_{2;\,h-\frac{1}{2}}^{n} + \frac{\Delta t}{(\Delta x)^2}D_{2;\,h-\frac{1}{2}}^{n} \\
& + \frac{\Delta t}{\Delta \alpha}\delta_{\alpha;\,k}^{n+1}F_{\alpha;\,k+\frac{1}{2}}^{n} + \frac{\Delta t}{(\Delta \alpha)^2}D_{\alpha;\,k+\frac{1}{2}}^{n} - \frac{\Delta t}{\Delta \alpha}\left(1 - \delta_{\alpha;\,k-1}^{n+1}\right)F_{\alpha;\,k-\frac{1}{2}}^{n} + \frac{\Delta t}{(\Delta \alpha)^2}D_{\alpha;\,k-\frac{1}{2}}^{n} \\
& + \frac{\Delta t}{\Delta \beta}\delta_{\beta;\,l}^{n+1}F_{\beta;\,l+\frac{1}{2}}^{n} + \frac{\Delta t}{(\Delta \beta)^2}D_{\beta;\,l+\frac{1}{2}}^{n} - \frac{\Delta t}{\Delta \beta}\left(1 - \delta_{\beta;\,l-1}^{n+1}\right)F_{\beta;\,l-\frac{1}{2}}^{n} + \frac{\Delta t}{(\Delta \beta)^2}D_{\beta;\,l-\frac{1}{2}}^{n}
\end{aligned}
\right\} P_{h,k,l}^{n+1} \\[6pt]
& + \left\{
\begin{aligned}
& \frac{\Delta t}{\Delta x}\left(1 - \delta_{1;\,h}^{n+1}\right)F_{1;\,h+\frac{1}{2}}^{n} - \frac{\Delta t}{(\Delta x)^2}D_{1;\,h+\frac{1}{2}}^{n} \\
& + \frac{\Delta t}{\Delta x}\left(1 - \delta_{2;\,h}^{n+1}\right)F_{2;\,h+\frac{1}{2}}^{n} - \frac{\Delta t}{(\Delta x)^2}D_{2;\,h+\frac{1}{2}}^{n}
\end{aligned}
\right\} P_{h+1,k,l}^{n+1} \\[6pt]
& + \left[ \frac{\Delta t}{\Delta \alpha}\left(1 - \delta_{\alpha;\,k}^{n+1}\right)F_{\alpha;\,k+\frac{1}{2}}^{n} - \frac{\Delta t}{(\Delta \alpha)^2}D_{\alpha;\,k+\frac{1}{2}}^{n} \right] P_{h,k+1,l}^{n+1} \\[6pt]
& + \left[ \frac{\Delta t}{\Delta \beta}\left(1 - \delta_{\beta;\,l}^{n+1}\right)F_{\beta;\,l+\frac{1}{2}}^{n} - \frac{\Delta t}{(\Delta \beta)^2}D_{\beta;\,l+\frac{1}{2}}^{n} \right] P_{h,k,l+1}^{n+1} \\[6pt]
& + \left\{
\begin{aligned}
& -\frac{\Delta t}{\Delta x}\delta_{1;\,h-1}^{n+1}F_{1;\,h-\frac{1}{2}}^{n} - \frac{\Delta t}{(\Delta x)^2}D_{1;\,h-\frac{1}{2}}^{n} \\
& -\frac{\Delta t}{\Delta x}\delta_{2;\,h-1}^{n+1}F_{2;\,h-\frac{1}{2}}^{n} - \frac{\Delta t}{(\Delta x)^2}D_{2;\,h-\frac{1}{2}}^{n}
\end{aligned}
\right\} P_{h-1,k,l}^{n+1} \\[6pt]
& + \left[ -\frac{\Delta t}{\Delta \alpha}\delta_{\alpha;\,k-1}^{n+1}F_{\alpha;\,k-\frac{1}{2}}^{n} - \frac{\Delta t}{(\Delta \alpha)^2}D_{\alpha;\,k-\frac{1}{2}}^{n} \right] P_{h,k-1,l}^{n+1} \\[6pt]
& + \left[ -\frac{\Delta t}{\Delta \beta}\delta_{\beta;\,l-1}^{n+1}F_{\beta;\,l-\frac{1}{2}}^{n} - \frac{\Delta t}{(\Delta \beta)^2}D_{\beta;\,l-\frac{1}{2}}^{n} \right] P_{h,k,l-1}^{n+1}
\end{aligned}
\tag{7}
$$

where $h$: 0, 1, 2, …, $N_H$ denotes the domain of $x$; $k$: 0, 1, 2, …, $N_K$ denotes the domain of $\alpha$; $l$: 0, 1, 2, …, $N_L$ denotes the domain of $\beta$; and $n$: 0, 1, 2, … denotes the domain of time $t$. The expression for the $\delta_\alpha$ parameter is given in Eq. (8) below, while the expressions for the other $\delta$ parameters can be written in a similar manner.

$$
\delta_{\alpha;\,k}^{n+1} = \frac{D_{\alpha;\,i,j,k+\frac{1}{2},l}^{n} - \left( D_{\alpha;\,i,j,k+\frac{1}{2},l}^{n} - \Delta\alpha F_{\alpha;\,i,j,k+\frac{1}{2},l}^{n} \right) \exp\left[ \Delta\alpha \dfrac{F_{\alpha;\,i,j,k+\frac{1}{2},l}^{n}}{D_{\alpha;\,i,j,k+\frac{1}{2},l}^{n}} \right]}{\Delta\alpha F_{\alpha;\,i,j,k+\frac{1}{2},l}^{n} \left\{ \exp\left[ \Delta\alpha \dfrac{F_{\alpha;\,i,j,k+\frac{1}{2},l}^{n}}{D_{\alpha;\,i,j,k+\frac{1}{2},l}^{n}} \right] - 1 \right\}}
\tag{8}
$$

Eq. (7) is, therefore, the discretized version of the FPE that was obtained from the proposed methodology. This equation can be used to solve the stochastic Saint-Venant equations with uncertain parameters. Implicitly solving Eq. (7)



involves computing the joint PDF of the state variables within the $x$–$\alpha$–$\beta$ domain, from which a quantitative probabilistic description of the ensemble behavior and variability can be determined for the hydrologic system represented by the stochastic Saint-Venant equations. In the following sections, Eq. (7) will be applied to a hypothetical application problem, and its performance will be measured by comparing its results against the results obtained by using the MC approach.

## 3 Application of the Monte Carlo approach and the new Fokker–Planck Equation methodology to a hydraulic routing problem

A hypothetical hydraulic routing problem will be the illustrative example which the new proposed methodology will be tested on. The hypothetical problem involves a river reach that is 2.70 km long, sloping at 0.0015 throughout the whole reach, having a rectangular cross section with a constant width of 6.1 m, and having no lateral inflow. Initially (at $t = 0$), the river is assumed to have a steady uniform flow of 15.5 m$^3$ s$^{-1}$ throughout the reach. At t > 0, the upstream flow increases linearly to reach 56 m$^3$ s$^{-1}$ at $t = 20$ min, then decreases linearly back to reach the initial flow value at $t = 60$ min, after which it remains at the same constant flow value for $t > 60$ min. If all parameters are assumed to be known, the described problem will be deterministic in nature. However, for the purpose of this study, the problem is made stochastic through the uncertainty in the Manning's roughness coefficient, whose statistical characteristics must be estimated.

### 3.1 Estimating the statistical characteristics of Manning's roughness coefficient

Manning's roughness coefficient ($n$) represents the resistance of the bed of a channel to the flow of water in it (Chow, 1959). Generally, a higher $n$ value defines a greater resistance to the flow. The value of this coefficient depends on several factors, including: surface roughness, vegetation, obstructions, as well as channel irregularity and alignment (Chow, 1959), all of which may exhibit considerable variability along the length of an open channel. It is, therefore, crucial to account for such variability in order to better represent the behavior of the unsteady open-channel flow system being solved.

The published and technical studies with sizeable datasets to address the variability of $n$ in such open channels are few. However, some studies have provided median and range values, while others have attempted to fit different probability distributions to the data (Gates and AlZahrani, 1996). Using such information, Manning's $n$ was assumed for this study to be a random variable with a normal distribution having a mean of 0.035 and a standard deviation of 0.005. The mean and standard deviation were chosen in a way that provides realistic values and distributions of $n$ that fall within the ranges and statistics provided by Gates and AlZahrani (1996) and that conform to the table of typical values in Chow (1959). Moreover, the generated $n$ values from the assumed statistics never fell below 0.01, thus complying with the fact that the roughness coefficients for flows in natural streams and excavated channels are always greater than 0.01 (Chow, 1959). As such, no truncation or discarding of any generated $n$ values was required. The chosen probability density function (PDF) for the roughness coefficient was used by the MC approach as well as by the new FPE methodology when solving for the ensemble behavior and variability of the hypothetical flow problem.



## 3.2 Application of the Monte Carlo approach

For the hypothetical routing problem of this study, the MC approach requires repeatedly solving the Saint-Venant equations in a deterministic manner for a large number of different roughness coefficient ($n$) realizations. To deterministically solve the Saint-Venant equations in their full form, several numerical techniques have been developed because their analytical solution has not been possible due to the presence of nonlinear terms (Sturm, 2001; Chaudhry, 2008). As such, Eqs. (1) to (4) were solved numerically through a finite-difference discretization, which is detailed in several references, e.g., Viessman et al. (1977) and Sturm (2001). The Courant condition was used in order to ensure the stability of the numerical method being used (Sturm, 2001). Furthermore, two boundary conditions, one at each end of the reach, and two initial conditions were defined in this study, since the problem deals with the subcritical unsteady non-uniform flow case (Sturm, 2001). As initial conditions, the discharge at every location along the river was provided (taken as 15.5 $m^3\,s^{-1}$, as explained in the problem description), and the flow was assumed to be initially uniform and steady. As for the boundary conditions, the flow hydrograph at the channel entrance was given, while at the downstream end, the channel was assumed to be hydraulically long so that the flow can be taken as normal flow, thus satisfying Manning's equation.

Following the preceding discussion, the Saint-Venant equations were deterministically solved for a total of 1000 times, each time using a different realization of $n$ that was generated based on the PDF chosen in Sect. 3.1. This MC approach provided a large number of solution realizations that formed an ensemble for each of the dependent variables, which were then analyzed in order to deduce their statistical properties, e.g., means and variances. The statistical results of the dependent variables are provided in Sect. 4, where they will be compared to the results of the new proposed FPE methodology.

## 3.3 Application of the proposed Fokker–Planck Equation methodology

Unlike the MC approach, the FPE methodology aims at solving for the ensemble behavior and variability of the stochastic system in one shot. The FPE methodology solves for the probability density function of the unsteady open-channel flow dependent variables over time and space, thus providing the ability to statistically describe the system in one simulation run.

Solving the hypothetical routing problem following the FPE methodology involved solving Eq. (7). Since this equation is the result of implicit discretization, it is unconditionally stable and requires no constraint on the size of the time step for its stability. However, to obtain sufficiently accurate solutions, the time step must still be limited with the Courant condition (Zheng and Bennett, 2002). Therefore, at every time position, the multidimensional Courant condition was checked in order to determine the appropriate size of the next time step.

Furthermore, to correctly solve the hydraulic routing problem, both the initial and boundary conditions of the problem in the physical space must be accurately represented in the probability space for use in the FPE methodology. Note that when the initial condition in the physical space is taken as deterministic, the Dirac delta function ($\delta(s)$) can be used to



represent it in the probability space. For the multidimensional case, the Dirac delta function can be written as a product of one-dimensional Dirac delta functions. Assuming that $H$ is a vector of $m$ state variables ($m$-dimensional), and $H_0$ is the corresponding vector of initial conditions, the initial condition PDF of $H$ can be written as

$$P(H, t) = P(H, 0) = \delta^m(H - H_0) \tag{9}$$

where $m$ is the number of dimensions. For the purpose of this study, $H$ corresponds to a point in the $x$–$\alpha$–$\beta$ domain, $H_0$ represents the initial condition that defines the deterministic flow at time $t = 0$ in the $x$–$\alpha$–$\beta$ domain, and $m$ is equal to 3. However, since it is not possible to numerically represent the Dirac delta function as a function with infinite value at only one position, it was estimated in the probability domain as a function with a very high value over a small bottom width, while preserving an area of unity.

Concerning the boundary conditions, they are usually divided for FPEs into two categories: accessible and inaccessible. Inaccessible boundaries are defined as those boundaries that could never be reached if the process starts from any interior point of the domain. On the other hand, for accessible boundaries, there is a positive probability that these boundaries will be reached from the interior of the domain within a finite amount of time (Feller, 1954). Accessible boundaries can be further subdivided into absorbing and reflecting, or no-flux, boundaries.

Note that since the FPE can be considered as the conservation equation for probability mass, a probability mass of unity needs to be conserved in the probability domain of the system. Therefore, using a reflecting (no-flux) boundary condition would be the most suitable choice for this study in order to ensure the completeness of the probability domain and to prevent any "particles" from leaving the domain (Gardiner, 1985). Such a condition was used to describe the boundaries of both the $\alpha$ and $\beta$ dimensions, noting that the minimum and maximum boundaries of both $\alpha$ and $\beta$ were chosen to be far enough so that they would encompass all possibilities that could occur for the considered routing problem. Moreover, recall that the upstream discharge hydrograph was assumed to be known. As a result, the probability densities at the upstream boundary of the $x$–dimension were known for all $t > 0$, whereas the downstream boundary in the $x$–dimension was extended enough so as to eliminate any of its effect on the numerical solution.

Therefore, following the application steps just described, the FPE methodology was applied to obtain the solution of the hypothetical stochastic open-channel flow problem and to determine the ensemble behavior of the system and the statistical distributions of the flow variables, as will be discussed in Sect. 4.

## 4 Numerical results and discussion

Both the MC approach and the proposed FPE methodology were used to solve the stochastic Saint-Venant equations of the hypothetical hydraulic routing problem presented in Sect. 3. While the state variables directly solved for were the velocity and depth (or celerity), the discharge was easily computed from these two variables.



A plot of the ensemble average discharge over time and space computed by the FPE methodology can be seen in Fig. 1 alongside a plot of the same results that were obtained by the MC simulations. From this figure, it is clear that the ensemble average discharge computed by the FPE methodology resembles the one obtained from the MC simulations quite well, while showing the same behavior and evolution of the mean discharge in both time and space as a result of the applied upstream wave. From the average behavior of the system, both plots show that the wave that was initiated upstream is observed to be transmitted downstream through the reach. In this case, the discharge at every location is seen to increase from the initial value of 15.5 $m^3$ $s^{-1}$ to some peak value, after which it decreases back again to the initial flow value. However, it is noticeable in both plots that the average peak discharge becomes lower at locations further downstream. This decrease comes as a result of the dissipation of energy through viscous effects as the water flows along the reach (Jeppson, 2011). In fact, the shearing stresses due to the vertical velocity gradients within the water lead to a loss of the fluid energy (mainly kinetic) as non-recoverable energy (e.g., increase in temperature), causing the peak velocity and discharge to decrease along the reach.

For a clearer comparison between the FPE and MC mean discharge results, cross sections of both plots from Fig. 1 at specific times and at specific channel locations were compared individually. The mean discharge was plotted as a function of location at different times (Fig. 2) and as a function of time at different locations (Fig. 3). From Fig. 2, it is clear that the change in the ensemble average discharge as a function of position (location within the channel reach along the flow direction) as computed by the FPE methodology almost perfectly replicates the corresponding results obtained by the MC simulations, with very minimal differences among the two. In both methods, the effect of the wave is clear as it causes an increase in the discharge first upstream, and then throughout the reach over time. Comparing the mean discharge at specific positions over time (Fig. 3) reveals that the FPE methodology also predicts the temporal change in the mean discharge very well at different channel locations. In fact, the timing and positions of the peak discharges predicted by the FPE methodology are very similar to those of the MC approach in all plots of Fig. 3, with a maximum relative difference of only around 6%. Therefore, it may be inferred from these results that the ensemble average discharge is predicted quite well by the FPE methodology when compared against the results obtained from the MC simulations.

Similar results were also plotted for the ensemble average flow depth. Figure 4 shows the mean river channel flow profile at different times as computed by the FPE and the MC methods. This figure shows how the applied upstream wave affects the depth profile of the river at different times, which is similarly predicted in both the FPE and MC simulations. Compared to the MC results, the ensemble average depth is predicted quite well by the FPE methodology, which only slightly overestimates the mean depth values at times between $t = 30$ min and $t = 45$ min. A comparison of the changes in depth as a function of time (Fig. 5) reveals that the FPE methodology provides a good match to the MC results while showing the same peaking pattern as the wave reaches a specific channel location. The timings of the peaks of the FPE methodology greatly match those of the MC simulations. However, similarly to Fig. 4, a slight overestimation can be noticed from the FPE methodology especially around the peak depths, but the maximum relative difference between both methods





was only around 7.5%. Hence, Figs. 4 and 5 both reveal that the FPE methodology can represent well the temporal and spatial evolution in the ensemble average flow depth due to the applied upstream wave, when compared to the MC results.

As for the velocity, Figs. 6 and 7 show the same two kinds of plots as before by comparing the mean ensemble average velocity results of the FPE methodology to those of the MC approach. Figure 6 shows that the mean velocity computed by the FPE methodology provides a good match to the MC results, with a slight underestimation. The pattern of change in the mean velocity as a function of position that is presented by the FPE methodology is very similar to that shown by the MC simulations, which demonstrates the effects of the applied upstream wave as it travels downstream with time. Figure 7 also reveals a good match between the two modeling approaches, with the FPE methodology providing the same pattern of velocity change over time as computed by the MC simulations. Similarly, a slight underestimation of the FPE mean velocity is also seen in Fig. 7, but the maximum absolute relative difference between both methods was only around 11%. Hence, in both figures, the FPE methodology seems to maintain its ability to provide a good match to the mean velocity values and patterns obtained by the MC simulations.

Therefore, from the preceding discussion, it may be inferred that the FPE methodology can predict the ensemble average discharge, depth, and velocity well over time and space even with the simplifications and assumptions applied when deriving the methodology. Since the ensemble average results generally match between the FPE and MC methods, and since the differences between them are mostly minute, these results provide support and validation to the simplifications and assumptions applied to the FPE methodology that was used to estimate the ensemble average behavior of the system.

Although checking the ensemble average of the flow variables provides an estimate of the mean behavior of the system, it is also important to check the variability of the system in order to have an idea about the range of possible results that may be expected. The standard deviation may be considered a good measure for such variability. Focusing on the flow discharge for the rest of the discussion, the standard deviation for the discharge computed by the FPE methodology was compared against the MC results to see the relative performance of the FPE methodology in predicting the system's variability.

Figures 8 and 9 compare plots of the standard deviation of the flow discharge as computed by the FPE and the MC methods. Similar to the ensemble average plots discussed previously, the former figure presents the results as a function of channel location at different times, while the latter presents them as a function of time at different channel locations. Figure 8 reveals how the movement of the wave in the downstream direction causes an increase in the standard deviation of the discharge, thus increasing the variability in the discharge. After the whole channel is affected by the moving wave, the variability is seen to be highest at the downstream end of the channel (e.g., Figs. 8b, 8c, and 8d). As the wave subsides, the standard deviation starts to decrease (Figs. 8c and 8d). Comparing the results of the FPE methodology to those of the MC approach shows that the FPE results correctly predict the patterns of change in the standard deviation as a function of position at each of the different times (Fig. 8). While the values of the standard deviations are slightly overestimated in some of the FPE results, especially at $t = 30$ min, the differences in the results are relatively small. Therefore, it may be inferred



that the FPE methodology provides an effective representation of the spatial changes in the standard deviation of the discharge, when compared to the results of the MC simulations.

Concerning the changes in the standard deviation of the discharge over time, Fig. 9 shows that the standard deviation forms an *M*-shaped pattern at all the plotted locations. For the MC results, the magnitudes of the standard deviation

forming this *M*-shaped pattern are observed to be generally increasing as a function of the distance away from the upstream boundary. It is noticeable that the results of the FPE methodology also predict an *M*-shaped pattern and an increase in the magnitude of the standard deviation of the discharge as a function of location. The timings, shapes, and magnitudes of the first peak for the MC results are generally matched well by the FPE results as shown by the plots of Fig. 9. Moreover, the locations and magnitudes of the second peak are well-represented by the FPE results in Figs. 9a and 9b. However,

differences arise in the position and magnitude of the second peak for locations further downstream (Figs. 9c and 9d). In fact, while moving further downstream, the second peak is observed to be underestimated by the FPE results and slightly shifted back in time when compared to the MC simulations. Moreover, the standard deviation following the second peak shows a faster decline in time for the FPE methodology than for the MC approach, while the local minimum between the two peaks is seen to be overestimated by the FPE results.

Several factors may have caused the discrepancies in the standard deviation of the discharge as computed by the FPE methodology. Among those factors may be the approximations and assumptions that were applied in order to simplify the FPE methodology to an easily solvable degree. These approximations have simplified the computations of the drift and diffusion coefficients of the equation, thus causing some discrepancies in the movement and the spread of the probability mass within the domain. Other factors may include the numerical method selected for computing the FPE methodology, as

well as the associated spatial and temporal discretizations that were used. All of these may have had an effect on the representation of the probability flow within the domain, thus leading to some discrepancies in the variability results.

Nonetheless, it may still be inferred that the FPE methodology performs satisfactorily in predicting the variability of the discharge in both space and time. In fact, the problem being solved is highly nonlinear, and involves a large uncertainty in one of its parameters. As such, estimations in the mean and variance are even more difficult to accurately quantify. The

25 FPE methodology was capable of not only correctly matching the general patterns of the spatial and temporal changes in the discharge standard deviation, but also correctly providing the standard deviation values within a range that is very similar to the range computed by the MC simulations. Moreover, these FPE results required a significantly less amount of time for computation as opposed to the MC results. In fact, the 1000 MC simulations ran for over 2 days, whereas the results of the FPE methodology were obtained in about 7 hours. Therefore, despite the uncertainty introduced by the roughness coefficient

and despite the approximations assumed for the FPE, the FPE methodology was able to provide an effective and satisfactory representation of both the spatial and temporal variability in the discharge within a much shorter simulation time period.

A final comparison between the FPE methodology and the MC approach is to compare the probability density functions (PDFs) of the flow discharge at different times and locations. The large number of simulations (1000) done using the MC approach provided an equal number of flow discharge results at each specific $x$–$t$ position in the space–time plane.



These values were then used in order to estimate the PDF of the discharge at that specific $x$–$t$ position. The FPE methodology, on the other hand, directly solved for the evolution of the PDFs of the state variables through time and space. However, recall that the FPE was solved in the $x$–$\alpha$–$\beta$ domain. With the discharge being a function of both $\alpha$ and $\beta$, the PDF of the discharge at each specific $x$–$t$ position was deduced from the $x$–$\alpha$–$\beta$ PDF provided by the FPE methodology.

For comparison, the PDFs of the flow discharge obtained from the FPE and MC methods were plotted at three different channel locations ($x$ = 900 m, 2250 m, and 2700 m) and at four different times for each location ($t$ = 15 min, 30 min, 45 min, and 60 min), as shown in Fig. 10. For most of the plots, the peak values of the PDFs predicted by the FPE methodology are similar to those computed by the MC approach. However, it may be noted that as the spread of the PDF becomes greater along the $x$-axis, the peak values become smaller in order to preserve an area of unity under the graph.

Therefore, a larger variability leads to lower PDF values. Such an example can be seen in Fig. 10g, where a major difference is noticed between the peaks of the PDFs resulting from the MC and FPE methods. While the peak of the MC PDF has a value of around 0.35, that of the FPE methodology has a value of around 0.065. The PDF of the FPE methodology has a larger spread, which causes the reduction in its PDF values. It may be noted that this plot corresponds to a point close to the local minimum of the plot in Fig. 9b, in which the FPE methodology is seen to predict a larger standard deviation than the

MC approach. Therefore, while both of the PDFs, corresponding to the FPE and MC approaches, seem to be located within the same range, thus providing similar expected values of the discharge, the greater variability of the FPE PDF causes its lower peak values.

      The prediction of the evolution of the PDFs for the flow discharge in space and time is a rather difficult task for the problem considered in this study. A lot of different factors are involved that would affect their evolution and calculation, 

including the step sizes in the $\alpha$ and $\beta$ directions whose values directly affect the computation of the discharge PDFs. However, the PDFs predicted by the FPE methodology are generally seen to be following similar trends to those computed by the MC approach (Fig. 10) while also satisfactorily predicting the ranges and locations of the PDFs (along the $x$-axis). Thus, with all the variability encompassing the routing problem considered, and with all the assumptions and approximations used during the application of the FPE methodology, the probability densities predicted by the proposed FPE methodology

are considered to be rather encouraging.

## 5 Summary and conclusions

      This study applied the proposed FPE methodology derived in the companion paper by Dib and Kavvas (2017) to a stochastic unsteady open-channel flow problem, with an uncertain roughness coefficient. The equations used to describe the open-channel flow problem were the Saint-Venant equations, transformed into their characteristic form by the method of 

characteristics. The proposed FPE methodology was applied in order to solve for the probability density of the flow state variables (velocity and depth/celerity, as well as discharge) and to provide a quantitative description of the expected behavior and variability of the stochastic system in one single simulation, as opposed to the large number of simulations usually





performed by the MC approach. The unsteady open-channel flow problem was also solved using the MC approach in order to use its results for evaluating the performance of the proposed FPE methodology.

Comparisons of the FPE results to those of the MC simulations revealed the effectiveness of the proposed FPE methodology in describing the ensemble behavior and variability of the stochastic Saint-Venant open-channel flow problem,
with an uncertainty in the roughness coefficient. In fact, the FPE methodology was found to replicate the ensemble average discharge of the MC simulations quite well in both space and time. In addition, it was also capable of effectively representing well the temporal and spatial change in the ensemble average depth and the ensemble average velocity. Furthermore, this method provided a good representation of the patterns and ranges of the standard deviation of the discharge over time and space, and showed a satisfactory prediction of the spatial and temporal trends and ranges of the flow
discharge PDFs. These encouraging results were obtained despite simplifications applied to the FPE methodology. Furthermore, with the FPE approach the simulation time period was significantly less than the time taken by the MC approach. Moreover, the FPE methodology results were obtained by running only one simulation, as opposed to the large number of simulations performed by the MC approach. Therefore, the results obtained in this study indicate that the proposed FPE methodology may be a powerful and time-efficient approach for predicting the ensemble average and variance
behavior, in both space and time, for the unsteady open-channel flow process under an uncertain roughness coefficient.

While this study considered the uncertainty in the system to be originating from the roughness coefficient, uncertainties may arise from other sources as well. Hence, future research could entail investigating the uncertainties due to the channel slope, channel cross section, lateral inflows, as well as initial and boundary conditions. Moreover, applying the proposed FPE methodology to systems which include more than one source of uncertainty could be a further extension of the
methodology in its attempt to effectively and efficiently describe such highly nonlinear and stochastic systems.

## 6 Data availability

This study involved the application of a new proposed methodology for the stochastic solution of a hypothetical unsteady open-channel flow problem. All the required parameters of this flow problem are provided in the study, and thus may be easily used for replicating the solution, if needed.

*Competing interests*. The authors declare that they have no conflict of interest.

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





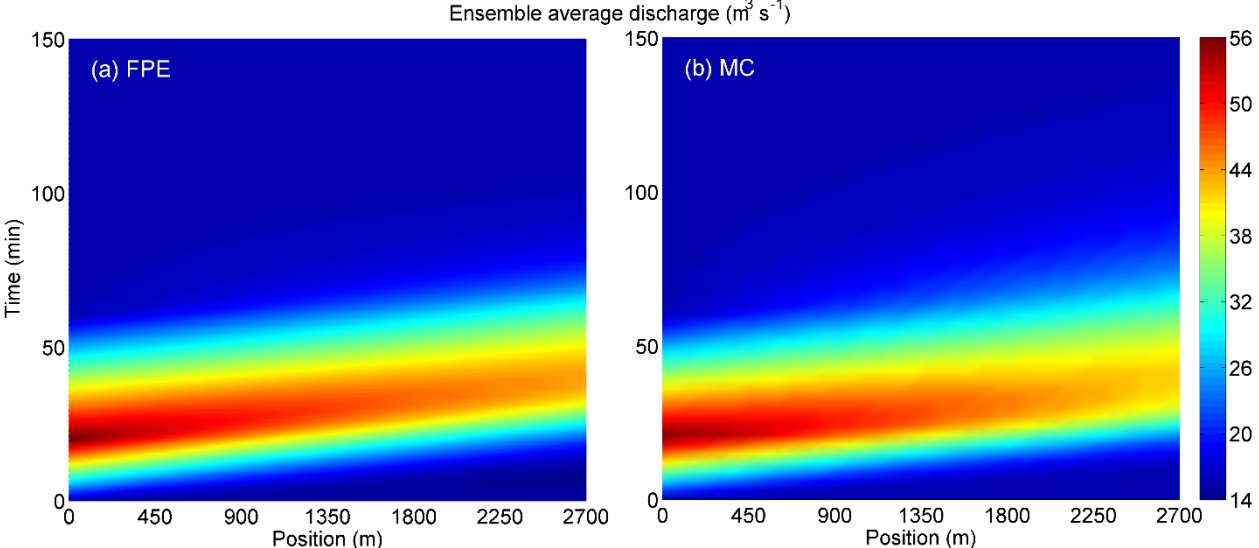

**Figure 1: Ensemble average discharge over channel position and time obtained by (a) the FPE methodology and (b) the MC approach.**



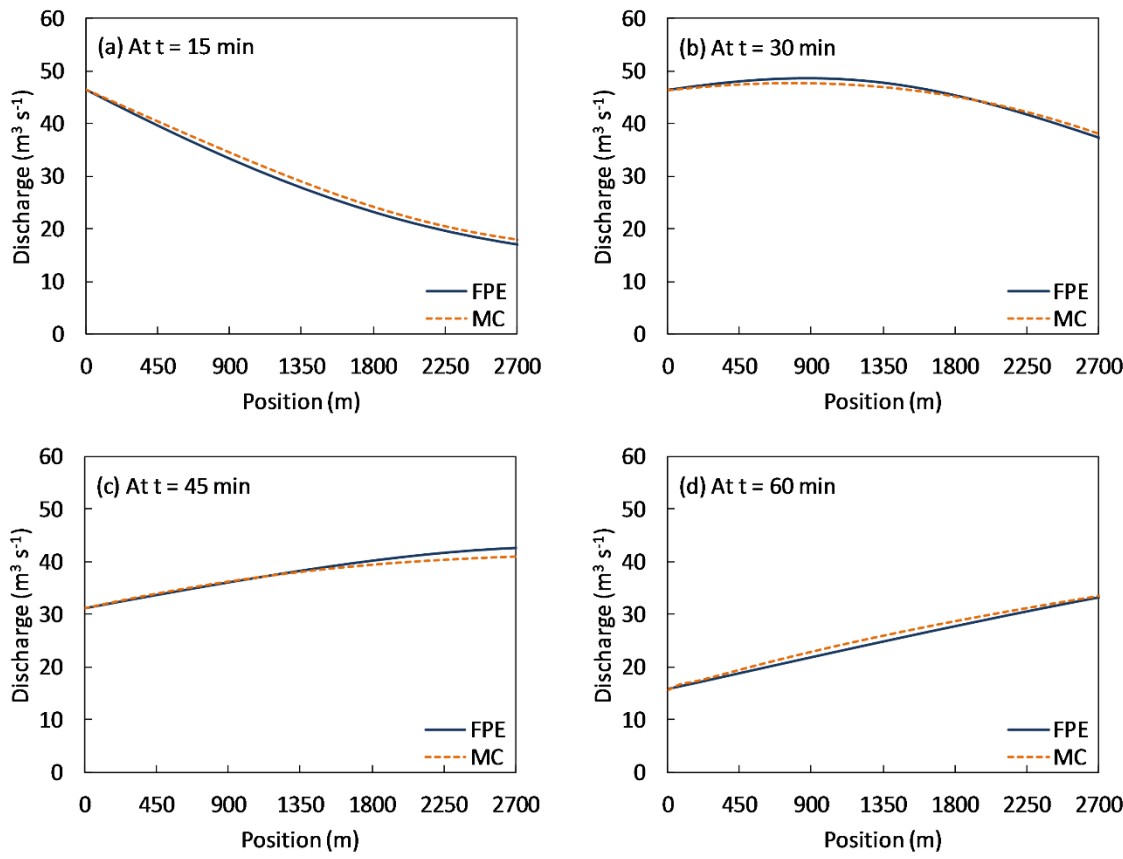

**Figure 2: Comparison of the ensemble average discharge obtained by the FPE methodology and the MC approach as a function of channel location (Position), at different times.**





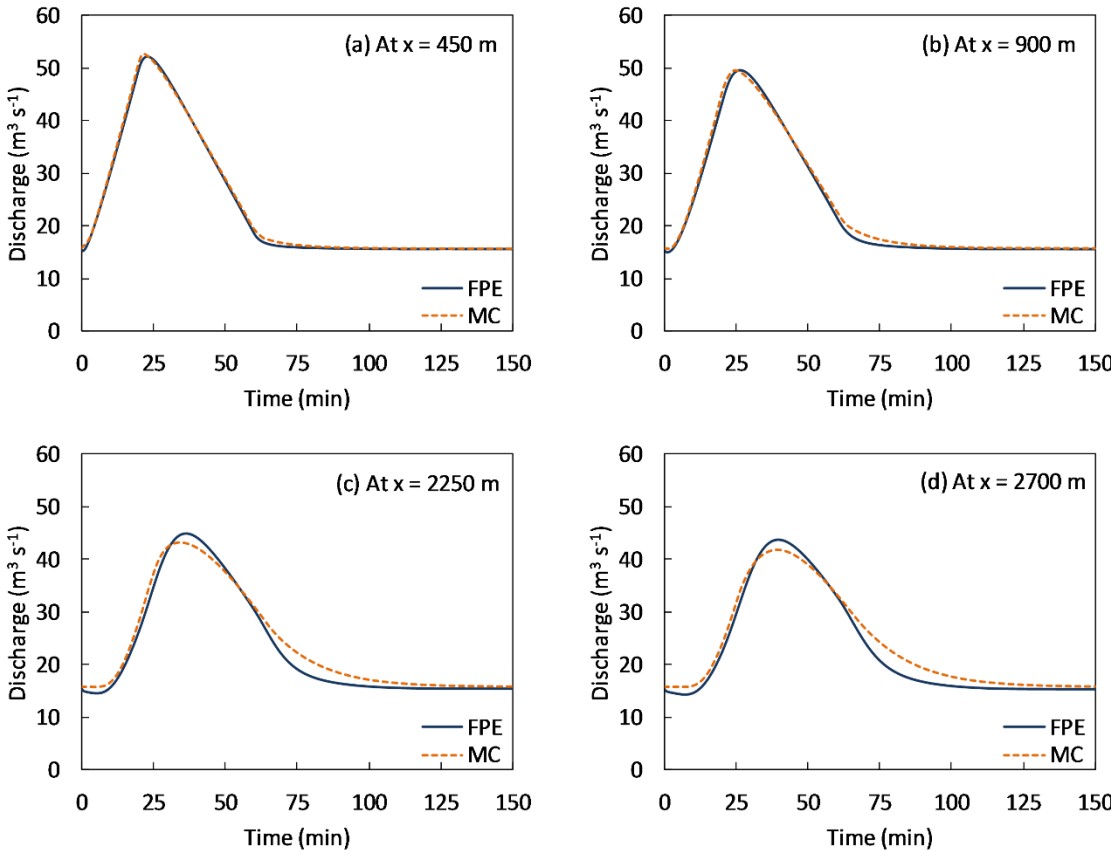

**Figure 3: Comparison of the ensemble average discharge obtained by the FPE methodology and the MC approach as a function of time, at different channel locations.**




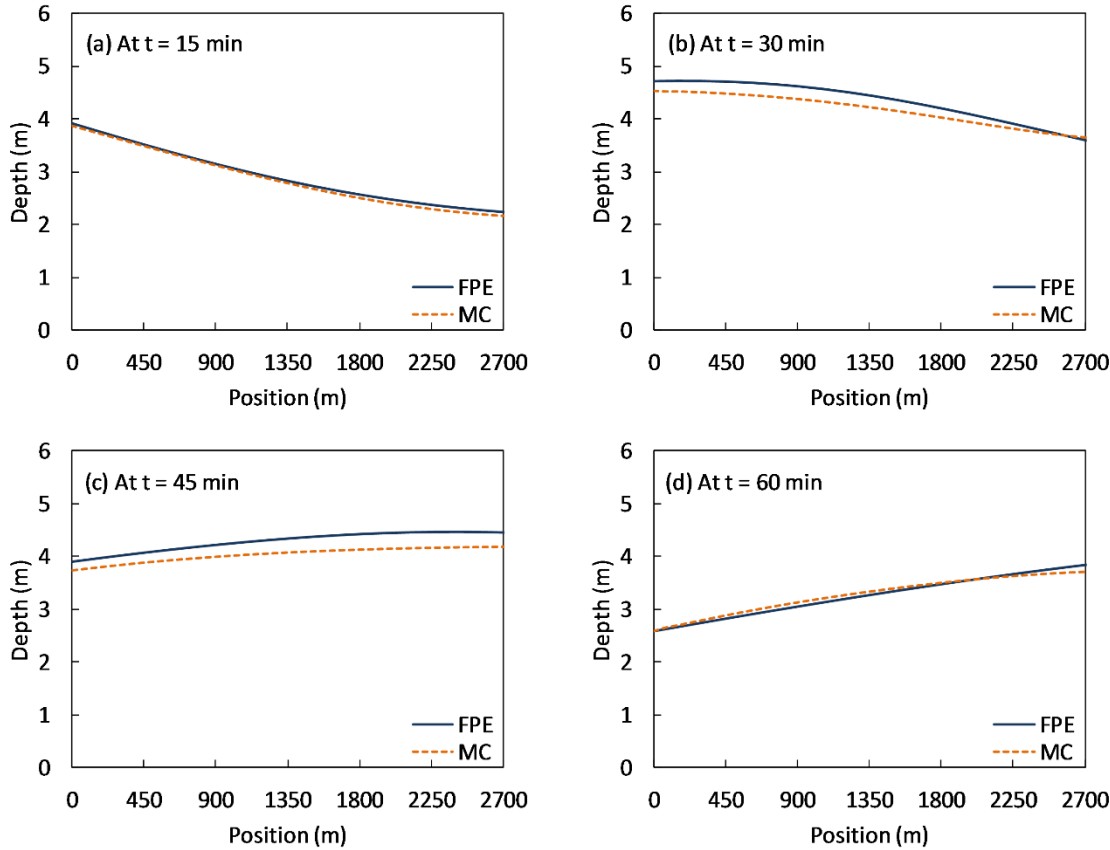

**Figure 4: Comparison of the ensemble average flow depth obtained by the FPE methodology and the MC approach as a function of channel location, at different times.**





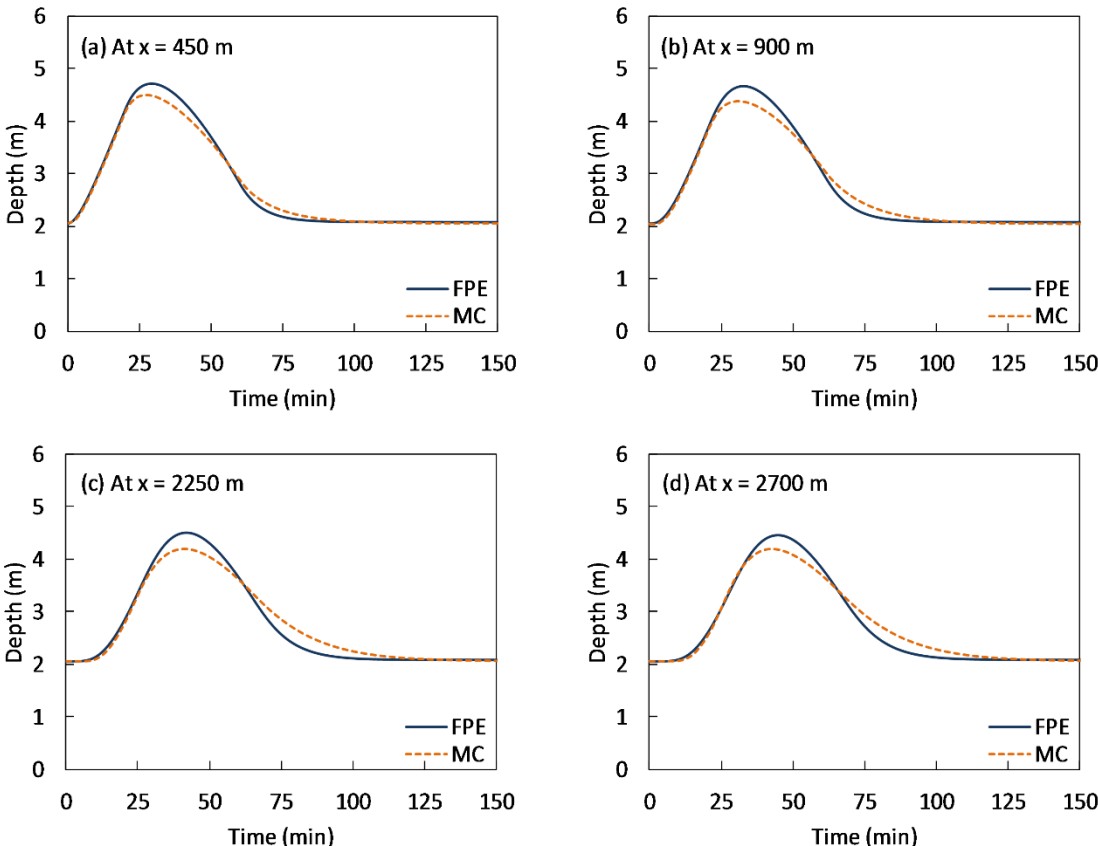

**Figure 5: Comparison of the ensemble average flow depth obtained by the FPE methodology and the MC approach as a function of time, at different channel locations.**




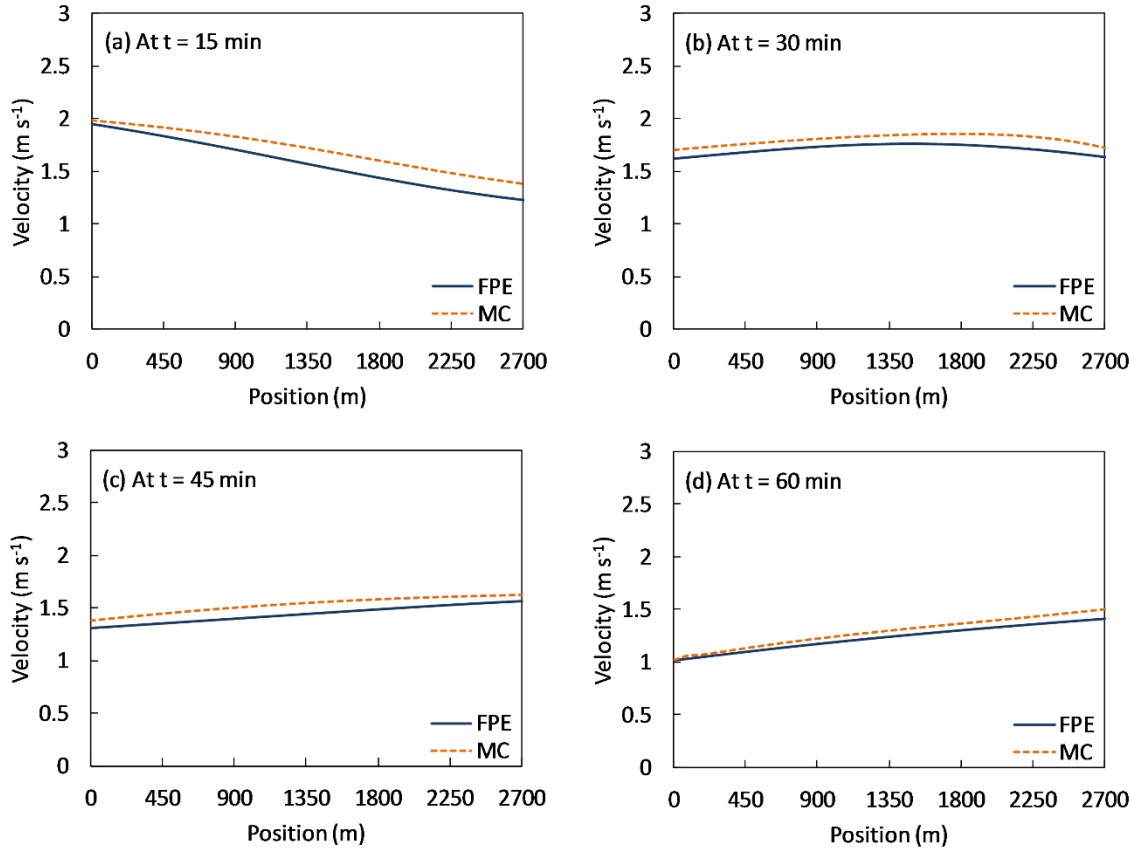

**Figure 6: Comparison of the ensemble average velocity obtained by the FPE methodology and the MC approach as a function of channel location, at different times.**



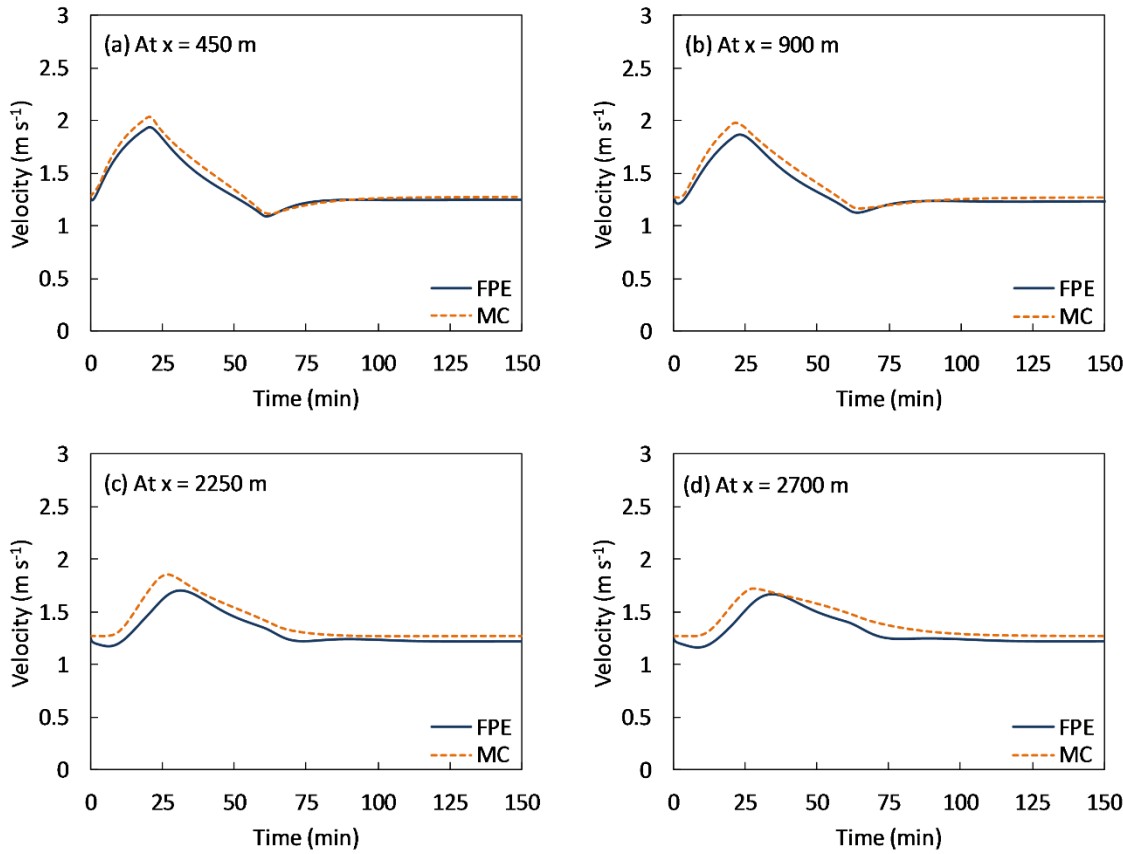

**Figure 7: Comparison of the ensemble average velocity obtained by the FPE methodology and the MC approach as a function of time, at different channel locations.**





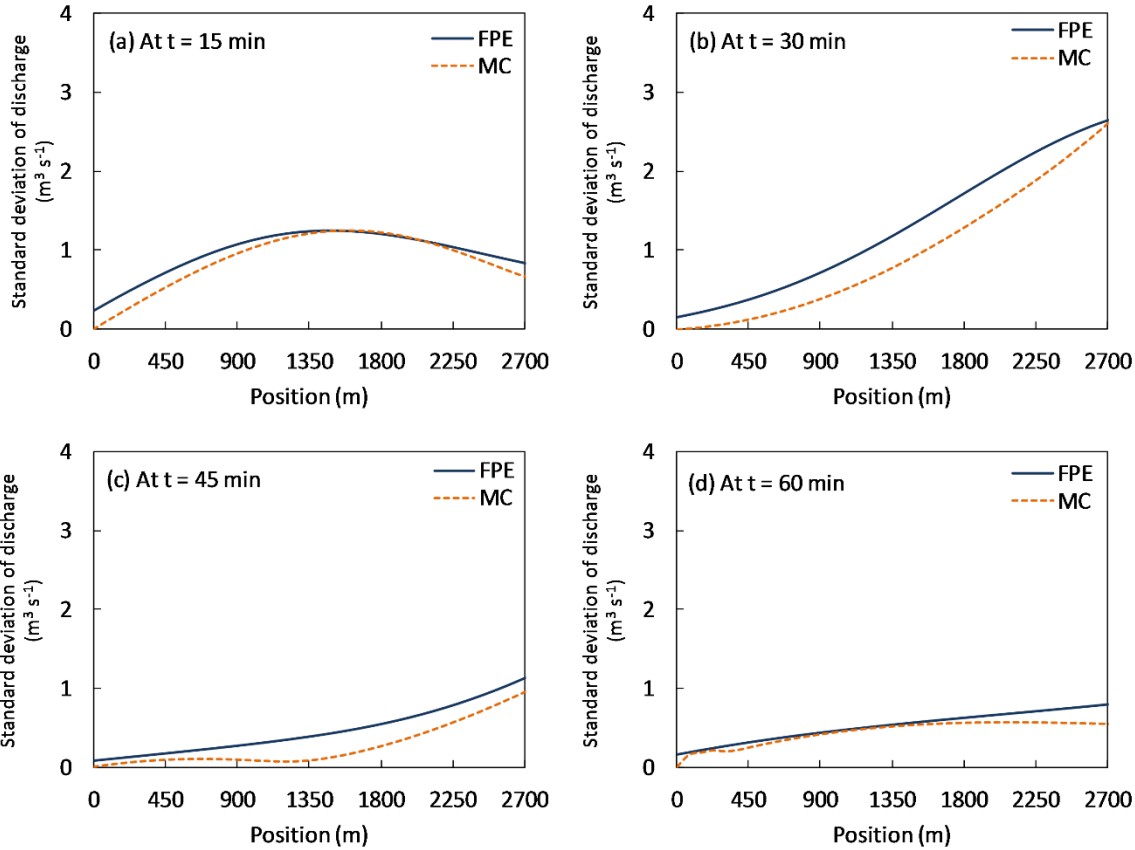

**Figure 8: Comparison of the standard deviation of the flow discharge obtained by the FPE methodology and the MC approach as a function of channel location, at different times.**





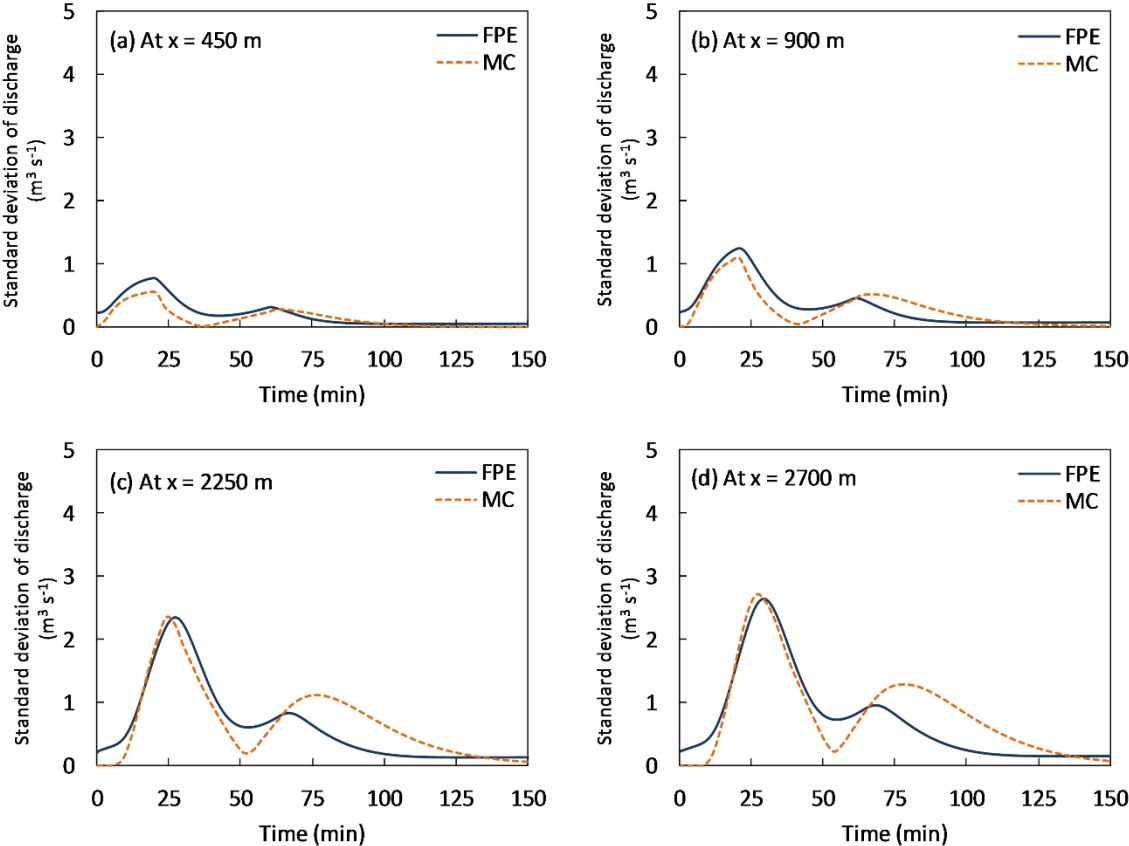

Figure 9: Comparison of the standard deviation of the flow discharge obtained by the FPE methodology and the MC approach as a function of time, at different channel locations.





**Figure 10: Comparison of the probability density functions of the flow discharge obtained by the FPE methodology and the MC approach, plotted at different times and channel locations.**