# Peer review of "Ensemble modeling of stochastic unsteady open-channel flow in terms of its time-space evolutionary probability distribution: numerical application"

_Hydrology and Earth System Sciences, 2017_

## Referee Comment (RC1) · Anonymous Referee #1 · 31 Aug 2017

General comments:

The paper compares the numerical results of PDFs of the state variables of an unsteady open-channel flow to Monte Carlo reference simulations. The PDF equations are derived in a companion paper. The mean discharge, the mean flow velocity, the mean flow depth, and the PDFs of the discharge are compared in detail.

The comparison of the numerical results of the method derived in the companion paper is of interest, but some important points are missing, which will be addressed in the

comments below.

Specific comments:

Please shorten your manuscript. It has too many repetitions and some statements are obvious.

In the abstract, on l. 1, you state that you use a "newly proposed Fokker-Planck Equation (FPE) methodology", whereas on p. 2, l. 32, you state that such a methodology has been applied many times. Please be more specific about what exactly is new about your methodology.

Please provide more information about the numerical setup of the MC simulations, like details about the finite-difference scheme, time step, grid size, parallelisation, ...

The paper will benefit from a plot of the standard deviation of the discharge dependent on the position and the time, analogue to figure 1. Please add such a plot.

Your choice for 1000 Monte Carlo realisations seems arbitrary. This is especially problematic when comparing the computational times of the MC simulations and the FPE simulations. Please add a comparison of MC simulations with fewer realisations, resulting in about 7 hours of computational time, like the FPE simulations, to the MC results with 1000 realisations. If one is only interested in the ensemble averages, it is very likely that fewer ensemble members will be sufficient for accurate results. This implies that you compare the results for the ensemble averages and also for the standard deviations with 1000 and fewer realisations. Maybe also add the results for 500 realisations. Furthermore, MC simulations are predestined for parallelisation. Neither did you write if the MC simulations where computed in parallel, nor did you incorporate this in your comparison of the computational times between the two different approaches. How difficult would it be to parallelise the numerical scheme of the FPE method?

On p. 11, l. 20 you write that the numerical errors caused by the spatial and temporal discretisation could lead to discrepancies in comparison to the MC results. Please

check this by performing simulations with higher and lower resolutions of the spatial and temporal discretisation.

Please comment on the implications for applications, like flood forecasting, of the errors made by the FPE approach.

On p. 13, l. 12-13 you state that it is an advantage that the simulation can be performed in only one run, but you do not motivate why this is an advantage. For parallelisation, it is even a disadvantage.

The discussion of the results lacks some points or is partly in contrast to what the figures show. Figures 3c,d show a slight decrease of discharge at early times with the FPE method, how do you explain that behaviour? Please explain the offset of the standard deviation at time t=0 for the FPE results. In figure 8c, the FPE result does not reproduce the decrease of the standard deviation. How do you explain the variation of the standard deviation at early times from the MC simulations in figure 8d? Figures 9 and 10 show more of a qualitative match of the results, than a quantitative match.

Your outlook on p. 13, l. 16-20 rather belongs to the companion paper. What about faster or more accurate numerical schemes? How could the discrepancies be reduced?

Technical corrections:

The tense of your abstract makes it read like a summary, please change the tense accordingly.

In the abstract you write that the total simulation period of the FPE method is smaller than that of the MC approach. You certainly mean computational time.

Check the indentations at the beginning of chapters and after equations, please delete them.

On p. 2, l. 31 you write that you do not limit the working space of the parameter

space. What about parameter combinations, where the neglected cross-covariance terms become large or the system shows a memory?

You write about the conservation of particles, although no particles where introduced in your paper, please reformulate.

On p. 2, l. 6 you write that parameters become random through uncertainties. You should write that the parameters are formulated as random functions in order to capture the uncertainties.

How did you ensure that Manning's coefficient never fell below 0.01, as described on p.6, l. 27?

The arguments of a PDF are usually separated by a semi-colon into arguments for which the PDF is a density and normal arguments, e.g. Pope (1985).

The notation of the m-dimensonal delta function is confusing, drop the exponent m.

P. 9, l. 11: What other form of energy, besides kinetic energy is dissipated due to shear stresses?

P. 9, l. 18: change "very minimal"

P. 13, l. 9: I do not think that the results for the PDFs are satisfactory in general. In some cases the are, in others they are not.

---

## Short Comment (SC1) · 12 Sep 2017

In this manuscript the authors present an illustrative example of the Fokker Planck equation associated to the De Saint Venant equation for a spatially uniform but random roughness coefficient. As I already commented in my review of the previous manuscript I think that the material is not enough to justify a two-paper series. Therefore, my suggestion is for merging the two manuscripts into one. The first part (up to page 5) of the second manuscript summaries what already presented in the first one and is not longer needed in the merged manuscript, while the second part can be easily merged

with the first manuscript.

In addition, I have the following concerns on to the illustrative example discussed in this second manuscript.

The boundary condition used at the lower end of the channel in the solution of the de Saint Venant Equation in a Monte Carlo framework are unclear and should be better specified and justified. At page 7 line 12 I read "... while at the downstream end, the channel was assumed to be hydraulically long so that the flow can be taken as normal flow, thus satisfying the Manning's equation". This sentence is unclear: what is normal flow? Mathematically the downstream condition can be of imposed water depth $y$, with the velocity $V$ obtained through the Manning's equation, or imposed velocity (or water discharge) and again the Manning's equation provides the water depth. The imposed condition can be either stationary, i.e. constant, or transient. It seems from the results that the authors choose the second option (Figures 2, 3 4 and 5 shows that the ensemble mean of both $y$ and $Q$ changes with time at the lower end of the channel), but no details on the specific boundary conditions are provided in the text. Assuming that one of the above two boundary conditions have been selected at the lower end of the channel, this choice should reflect to the boundary condition for the stochastic variables in the FPE equation. For instance, if $y$ is imposed its PDF is a Dirac delta, while the pdf of the velocity is related to the pdf of the roughness coefficient through the Manning's equation. In turn, the PDFs of $\alpha$ and $\beta$ depends from the PDFs of $y$ and $V$ through equations (13) and (14) of the first manuscript. Similar arguments can be applied if the BC is of imposed water discharge (i.e. velocity) and the water depth is computed through the Manning's equation. What puzzles me is that the authors impose a reflection boundary at the end of the channel (line 15, page 8), which is apparently not compatible with the previous conditions and with those imposed in the MC simulations. Since Figures from 2 to 5 show clearly that the water wave interacted with the downstream boundary, I am expecting that the boundary condition here should have an impact on the solution. This reinforce the need to select compatible boundary

conditions in both models. In addition, given the boundary condition of the FPE at the initial section I was expecting here the standard deviation of the water discharge equal to zero, as in the MC simulations. However, this is not the case, as shown in Figures 8 and 9. This unexpected result needs justification.

─────────────────────────

---

## Referee Comment (RC2) · Anonymous Referee #2 · 5 Jan 2018

The authors present a novel way of using the Fokker–Planck Equation (FPE) to derive directly (one simulation) the probability distributions of velocity and depth resulting from uncertain roughness in a hypothetical unsteady open-channel flow problem. Although the efficiency gain over Monte Carlo simulation for the particular case presented seems limited, improving direct ways for probabilistic modelling is a relevant contribution.

The paper is well-written and well-structured, with sometimes a bit too many reminders of the story-line and mentioning in an early stage already the main conclusions (e.g. P.2 l.28).

[Figure]

Main comments are: The title is confusing because of the "Ensemble modeling", whereas the main objective of the authors is to present a single simulation solution for providing a pdf. I suggest changing the title of this paper and the companion paper, taking out the term "Ensemble". (e.g. into something like "Fokker-Planck modelling of stochastic open-channel flow in term..", or "Deterministic modelling of..")

I would suggest to continue reporting and discussing the results for velocity and depth also in the latter part of the Results section (even if only in text, because with figures it would become too long), rather than only discussing discharge results. For velocity and depth, differences are likely to be larger and may lead to better understanding of what are the causes, because in discharge differences in velocity and depth may be cancelled out.

Please include a sensitivity analysis of the MC results with respect to the number of iterations. It would be interesting to check if with more simulations the results go nearer to the FPE result or further away (or no difference), and if with fewer simulations the same result is achieved. This is relevant for the claim on computational efficiency, as also pointed out by Referee#1 (fifth specific comment).

The analysis and discussion on computational time needs to be more detailed (including computational times, hardware used, etc.) and expanded. In particular with whether the FPE approach is suitable for parallelisation, if not, the MC analysis, for the case study presented, can be easily made more efficient. The authors could perhaps also include their expectations on the applicability and computational efficiency of their FPE method for larger systems. Would the gain with respect to MC increase or not?

The gain in computational efficiency, as presently described, seems limited. Hence, the claimed contribution there, in Abstract and Conclusions, should be down-sized or contextualised.

Detailed comments are: P.2 l.28: "..producing the complete ensemble model results.." is not correct, because, if I understand correctly, the method does not reproduce the

individual traces (ensemble members). Therefore, this should be something like "..producing the statistical properties.." P.7 l.14-15: Explain the choice of 1000 simulations. Report the sensitivity of the statistical characteristics to the number of simulations in the MC. P.8 l.27-28: Repetition. There is already a sentence connecting Sections 3 and 4 in lines 23-25. Consider leaving out one of the two. P.9 l.18: Repetition. Delete ".., with very minimal differences among the two" P.9 l.32-33: However, ... ,but... Consider reformulating. P.10 l.13-17: Reformulate removing redundancies. (Or consider leaving out, because it reads perhaps too much as general conclusions, while this is in the middle of presenting and discussing results) P.10 l.20: Why do the authors continue only with Discharge? Differences in velocity and depth may be cancelling each other in the resulting discharge. Also when thinking of flood risk management applications, it may be more interesting to look at velocity and depth variance. P.10 l.18-23: Too much repetition. Suggest to shorten and merge with next paragraph where actually the presentation of variability results starts. P.10 l.33: "relatively small" Suggest to add some of the differences in %. Also provide differences in standard deviation for velocity and depth. P.11 l.15-21: The results for velocity and depth may help in understanding the causes of differences in variability. P.11 l.27-31: As described in main comments above, please expand the analysis and discussion of computational efficiency, and make it a separate paragraph. P.12 l.18: General sentence. Consider deleting.

---

## Author Comment (AC1) · 1 Feb 2018

**Response to the comments of Anonymous Referee #1 published on August 31, 2017 concerning the manuscript with reference number: hess-2017-394.**

We would like to thank Referee #1 for his/her insightful feedback. Our responses to the specific points raised by the referee are provided below. Please note that the referee's comments will be presented in italics, preceded by a "**C**", while the corresponding authors' responses will be presented in normal typeface with a blue font, preceded by an "**R**". For some responses, the text which was changed or added to the manuscript (based on suggestions from the referee comments) is quoted and placed under "Specific author changes". Please note that the pages and line numbers provided in this document are from the original version of the manuscript.

**General Comments**

**C1:** *The paper compares the numerical results of PDFs of the state variables of an unsteady open-channel flow to Monte Carlo reference simulations. The PDF equations are derived in a companion paper. The mean discharge, the mean flow velocity, the mean flow depth, and the PDFs of the discharge are compared in detail.*
*The comparison of the numerical results of the method derived in the companion paper is of interest, but some important points are missing, which will be addressed in the comments below.*

**R1:** We would like to thank the referee for their review, and we will be addressing the specific points in our responses below.

**Specific Comments**

**C2:** *Please shorten your manuscript. It has too many repetitions and some statements are obvious.*

**R2:** Following the referee's suggestion, we have worked on adjusting the manuscript to remove any repetitions and obvious statements. On the other hand, we would like to note that we have added some text in several places of the manuscript as a response to some of the comments from the two Referee Comments and the Short Comment posted for this manuscript.

**C3:** *In the abstract, on l. 1, you state that you use a "newly proposed Fokker-Planck Equation (FPE) methodology", whereas on p. 2, l. 32, you state that such a methodology has been applied many times. Please be more specific about what exactly is new about your methodology.*

**R3:** After rereading the abstract and the introduction with the referee's point in mind, we understand how the writing may have been unclear concerning the novelty of the methodology being used. In fact, what was meant by the "methodology" mentioned on Page 2: Line 32 (of the original manuscript) was the "technique" developed by Kavvas (2003). This technique has been applied to other processes with different governing equations where FPEs specific to those processes were obtained and applied successfully. However, to the authors' knowledge, the technique had never been applied to the Saint-Venant equations to tackle the unsteady open-channel flow problem. As such, the novelty of the proposed FPE methodology that was developed in the companion paper was to figure out how to apply the Kavvas (2003) technique to the Saint-Venant equations, and then to go forth with developing the

FPE that is specifically for the stochastic unsteady open-channel flow process, which has not been developed before. As such, changes have been made to the abstract as well as to the second half of the introduction in order to clarify this matter to the reader.

Specific author changes
In the abstract, the second sentence now reads as follows

*"This methodology computes the ensemble behavior and variability of the unsteady open-channel flow process by directly solving for its time-space evolutionary probability distribution."*

The second half of the introduction has been considerably adjusted, and now reads as follows

*"    In order to circumvent having to solve the Saint-Venant equations repeatedly for a large number of times, this study uses a new methodology that solves for the time-space evolutionary probability distribution of the unsteady open-channel flow process in only one simulation. From this probabilistic solution, one can then obtain the ensemble mean and ensemble variance of the process as they evolve in time and space. This new methodology is proposed, explained, and derived in the companion paper by Dib and Kavvas (2017), which makes use of the ensemble averaging technique developed in Kavvas (2003) to obtain a Fokker–Planck Equation (FPE) that specifically describes an unsteady open-channel flow process. Some other hydrologic processes have been successfully simulated by following a similar procedure, which involved applying the ensemble averaging technique of Kavvas (2003) to their corresponding governing equations and obtaining the FPEs specific to their case. These include: unsaturated water flow (Kim et al., 2005a), root-water uptake (Kim et al., 2005b), solute transport (Liang and Kavvas, 2008), snow accumulation and melt (Ohara et al., 2008), unconfined groundwater flow (Cayar and Kavvas, 2009a, b), as well as kinematic open-channel flow (Ercan and Kavvas, 2012a, b).*

*    Note that in addition to producing the statistical properties in a computationally efficient manner through one simulation, the FPE methodology developed for the unsteady open-channel flow process directly solves for, and is linear in, the probability density of the dependent variables. Moreover, while this methodology assumes a finite correlation time for the considered process, it does not make any linearization assumptions and it does not have limitations on the working range of the parameter space.*

*    Therefore, in the wake of the preceding discussions, the first objective of this study is to use the FPE methodology derived in the companion paper for the unsteady open-channel flow process (Dib and Kavvas, 2017) and to apply it to a representative stochastic unsteady open-channel flow problem in order to solve for the probability density of the state variables of the flow process and to provide a quantitative description of the expected behavior and variability of the system in one single simulation. The second objective is to evaluate the performance of the proposed methodology and to validate its results by comparing the statistical properties of the flow variables computed by the FPE methodology against those calculated by the MC approach."*

**C4:** *Please provide more information about the numerical setup of the MC simulations, like details about the finite-difference scheme, time step, grid size, parallelisation, ...*

**R4:** Following the referee's suggestion, additional information has been provided regarding the MC simulations.

Specific author changes
Technical information has been added directly before Section 3.1 (on Page 6: Line 14):

*"Note that all simulations for the MC approach and the FPE methodology were run on a computer having 16 GB of RAM and an Intel i7 processor with four cores, each core having a base frequency of 2.40 GHz and a maximum frequency of 3.40 GHz."*

The below information has been added at the beginning of Page 7: Line 6:

*"For this study, the characteristic form of the Saint-Venant equations was discretized in an explicit manner by substituting the time derivatives with their first order finite-difference forms, as detailed in several references, e.g., Viessman et al. (1977) and Sturm (2001). The values of the dependent variables at the new time steps were computed at the points of intersection of the positive and negative characteristic curves, which rendered the final solution on an irregular x–t grid. These computations were parallelized and run over all four available cores (with no hyperthreading). The solution was then interpolated onto a rectangular grid, with a Δx of 75 m and a Δt of 3 min, by using a parallelized process which optimized the computational time by running the simulations over the all four available cores."*

**C5:** *The paper will benefit from a plot of the standard deviation of the discharge dependent on the position and the time, analogue to figure 1. Please add such a plot*

**R5:** The referee's suggestion was followed as detailed below.

Specific author changes
The requested plot has been added to the manuscript and denoted as Figure 8, and it is as follows:

[Figure]

*Figure 1. Standard deviation of flow discharge over channel position and time obtained by (a) the FPE methodology and (b) the MC approach.*

Moreover, a discussion regarding this figure has been added in the Discussion section on Page 10: Line 23 as follows:

*"Fig. 8 shows a comparison of the standard deviation of the flow discharge over space and time as computed by the FPE methodology and by the MC simulations. Both plots of this figure reveal that the standard deviation experiences two triangular areas of high values, the earlier in time being generally higher than the later, and both areas showing a stronger intensity further downstream. While the general resemblance of the FPE plot to the MC plot is good, the second area of elevated standard deviations is more compact in the FPE results and shows slightly lower values than those of the MC results. In an attempt to study such differences more closely, cross sections of both plots from Fig. 8 at specific times and at specific channel locations were compared individually (Figs. 9 and 10), in a manner similar to the ensemble average plots discussed previously."*

**C6:** *Your choice for 1000 Monte Carlo realisations seems arbitrary. This is especially problematic when comparing the computational times of the MC simulations and the FPE simulations. Please add a comparison of MC simulations with fewer realisations, resulting in about 7 hours of computational time, like the FPE simulations, to the MC results with 1000 realisations. If one is only interested in the ensemble averages, it is very likely that fewer ensemble members will be sufficient for accurate results. This implies that you compare the results for the ensemble averages and also for the standard deviations with 1000 and fewer realisations. Maybe also add the results for 500 realisations. Furthermore, MC simulations are predestined for parallelisation. Neither did you write if the MC simulations where computed in parallel, nor did you incorporate this in your comparison of the computational times between the two different approaches. How difficult would it be to parallelise the numerical scheme of the FPE method?*

**R6:** While it is true that a lower number of Monte Carlo (MC) realizations may be sufficient for accurately calculating the ensemble average of the process, this study is concerned with comparing not only the ensemble averages but also the standard deviations in order to determine the variability of the system considered. As such, the number of MC realizations must be large enough to numerically approximate, with sufficient accuracy, the second moment of the stochastic quantities for the problem at hand.

To discuss this issue, and following the referee's comment above, we ran the MC simulations with 50, 100, 200, and 500 realizations and we compared the flow discharge ensemble averages and standard deviations against those obtained from our 1000-realization run used in the manuscript. We plotted the percent relative differences of these simulations as compared to the results obtained from the 1000 realizations (Figure 2 and Figure 3 below).

From Figure 2, it is clear that the number of MC realizations affects the accuracy of the computed ensemble average flow discharge, but not too significantly (among the realizations shown). While increasing the number of realizations provides greater agreement with the results of the 1000-realization run, the relative difference was still low, even for the 50-realization run (a maximum of around 2.8%). Therefore, the number of realizations required for comparing the ensemble averages can be much smaller than 1000, with not much loss of accuracy.

However, the same cannot be said regarding the standard deviations of the flow discharge. In fact, it is very clear in Figure 3 that the number of realizations is extremely important in determining the accuracy

of the standard deviation results. When compared to the 1000-realization run, the standard deviations show absolute differences that reach or exceed 65% for 50 realizations, 35% for 100 realizations, 20% for 200 realizations, and 15% for 500 realizations. Therefore, while 1000 MC realizations may seem like a large number at first, the comparisons here show that this number of realizations is necessary to produce a sufficiently accurate computation of the standard deviation for the problem at hand.

As for the referee's comment regarding parallelization, we would like to note that the 1000-realization MC run simulated for this study was parallelized and run over four cores (with no hyperthreading), thus noticeably reducing the computational time as compared to an un-parallelized run. With such parallelization, the timing of the MC run with 1000 realizations was almost around 2.5 days, as stated in the manuscript. Using the same parallelization, we found that a MC run with 100 realizations took around 7 hours, which is similar to the time taken by the FPE methodology. However, as seen in the previous discussion, 100 realizations are not enough to provide sufficiently accurate results for the standard deviations, and thus we believe it would not be of much benefit to add these results to the comparison against the FPE methodology. Moreover, including in the manuscript such a comparison, as well as a comparison against 500 realizations, may cause a digression from the main idea of the manuscript, which is to gauge the performance of the FPE methodology. Therefore, we believe it would be preferable not to add such comparisons; however, we added some text informing the reader of the parallelization of the MC simulations, as we detailed in our reply **R4** above, as well as to clarify our choice of 1000 simulations in the manuscript (the added paragraph is quoted at the end of this reply).

Finally, concerning parallelizing the numerical scheme of the FPE methodology, the following may be said. If we observe the computational times of the implicit numerical solution of the FPE methodology, the portion of the code requiring the largest time turns out to be the filling out of the coefficient matrix, especially when the discretizations for $\alpha$ and $\beta$ are small. Therefore, parallelizing this portion of the code may allow one to reduce the computational time of this method. We believe that the difficulty in such parallelization is not too high. After deciding on the number of cores over which the parallelization will occur, one can adjust the code portion containing the loops that fill out the coefficient matrix in a way that divides the loops onto the different number of cores. As such, the coefficient matrix will be filled out in a shorter period of time.

[Figure]

*Figure 2. Percent relative difference for the flow discharge ensemble average obtained from MC simulations with 50, 100, 200, and 500 realizations, when compared to the results obtained from 1000 realizations.*

[Figure]

*Figure 3. Percent relative difference for the flow discharge standard deviation obtained from MC simulations with 50, 100, 200, and 500 realizations, when compared to the results obtained from 1000 realizations.*

Text was added to clarify further the choice of 1000 MC simulations by adjusting the paragraph on Page 7: Lines 14-19 to read as follows:

"     *Following the preceding discussion, the Saint-Venant equations were deterministically solved for a total of 1000 times, each time using a different realization of n that was generated based on the PDF chosen in Sect. 3.1. While a lower number of realizations may have been sufficient for accurate computations of the first moment, it would not have been sufficient for the accurate computations of the second moment. In fact, the standard deviation of the flow discharge computed using 50, 100, 200, and 500 realizations showed absolute relative differences that reached 65%, 35%, 20%, and 15%, respectively, when compared to the results of the 1000-realization run. Therefore, the number of realizations in this study was selected to be large enough to numerically approximate, with sufficient accuracy, both the first and the second moments of the stochastic quantities of the problem at hand.*"

**C7:** *On p. 11, l. 20 you write that the numerical errors caused by the spatial and temporal discretisation could lead to discrepancies in comparison to the MC results. Please check this by performing simulations with higher and lower resolutions of the spatial and temporal discretisation.*

**R7:** With the Courant condition being used to limit the time step in the FPE runs, the temporal discretization becomes variable during a simulation, while also being affected by the α-β discretization. So there is no user-decided constant time step for the FPE simulation. However, the main effect that was alluded to by the phrase quoted by the referee is the one occurring due to the discretization in the α and β dimensions. This is because using relatively large α and β discretizations may cause discrepancies in the FPE results. Therefore, following the referee's suggestion, we ran the FPE methodology using several discretizations to test what differences may incur. The discretization in the α-β plane used for the results in the manuscripts was $\Delta\alpha = \Delta\beta = 0.5$, and will be presented as 0.5x0.5 ($\Delta\alpha$x$\Delta\beta$). The additional FPE runs were done with several discretizations, some with higher and some with lower resolutions. Sample results of these are shown for the discharge and its standard deviation in Figure 4 and Figure 5, respectively.

In both of these figures, the subplots on the left show the plots of the discharge or standard deviation of the discharge of all the different discretizations attempted, along with the MC results (thick red line). Whereas, the subplots on the right show only the few high-resolution discretizations that are close to the one used in the manuscript.

Looking at both figures, one can notice a clear difference in the results when the FPE methodology is run with lower resolutions. In fact, such a difference is clear in the discharge results (Figure 4, left subplots) and is even more prominent in the standard deviation results (Figure 5, left subplots). However, the subplots on the right show that the higher resolutions provide plots that converge to the same locations, and that are the closest to the MC results. These resolutions included the one used in the manuscript (0.5x0.5) as well as 0.25x0.25, 0.33x0.33 and 1x1, all of which provide similar results. As such, from these figures, it is clear that the resolution used in this study was high enough not to incur any major discrepancies in the FPE results, but that there is a high possibility for such discrepancies to occur if lower resolutions are used.

[Figure]

*Figure 4. Plots of discharge results obtained from the FPE methodology solved using different α-β discretizations. The results of the Monte Carlo (MC) simulations are also plotted for comparison (thick red line). The figures on the left show all the different discretizations attempted; the figures on the right show only the few discretizations closest to the one used in the manuscript (0.5x0.5).*

[Figure]

*Figure 5. Plots of the standard deviation of discharge results obtained from the FPE methodology solved using different α-β discretizations. The results of the Monte Carlo (MC) simulations are also plotted for comparison (thick red line). The figures on the left show all the different discretizations attempted; the figures on the right show only the few discretizations closest to the one used in the manuscript (0.5x0.5).*

**C8:** *Please comment on the implications for applications, like flood forecasting, of the errors made by the FPE approach.*

**R8:** Following the referee's comment, we comment on this issue as detailed below.

Specific author changes
Text was added to the beginning of the paragraph starting on Page 11: Line 22 as follows:

> *"Such discrepancies may be faced when using the FPE methodology in engineering flow applications, such as flood forecasting and flood control. The variability of the flow in flood forecasting applications, for example, may be underestimated at the downstream end of the reach, specifically during the later time periods. This would impact the range of flows that are forecasted to occur at the downstream end."*

**C9:** *On p. 13, l. 12-13 you state that it is an advantage that the simulation can be performed in only one run, but you do not motivate why this is an advantage. For parallelisation, it is even a disadvantage.*

**R9:** As described in the manuscript, one of the advantages of having the simulation performed in only one run is the reduced computational time and expense required as compared to the MC simulations. However, while such an advantage may not seem immense when only one uncertain parameter is involved (especially with the possibility of parallelizing the MC simulations), the advantage becomes much more prominent when the governing equations involve a larger number of uncertain parameters and boundary conditions. In this case, the computational expense of MC simulations would exponentially increase due to the higher number of simulations needed to maintain the desired accuracy in the results, thus significantly increasing the computational time regardless of parallelization. On the other hand, the FPE methodology would only require simple adjustments of the FPE in order to account for the additional uncertainties. After that, the FPE would be solved in the same way as was done for this study, with barely any implications on the computational expense (more details regarding possible changes to the FPE in such a situation are provided in our reply to **C4** of Referee #1 for the companion paper). The advantage is made clearer to the reader in the Conclusion as detailed below.

Specific author changes
The second half of the second paragraph of the Conclusion section (Page 13: Lines 11-15) was adjusted as follows:

> *"Moreover, the FPE methodology results were obtained by running only one simulation, as opposed to the large number of simulations performed by the MC approach. Such an advantage becomes prominent with a greater number of uncertain parameters and boundary conditions, in which case the computational expense of the MC simulations that is needed to preserve the desired accuracy would exponentially increase. On the other hand, only simple adjustments would be required for the FPE, which could then be solved as was done in this study, with minor implications on its computational expense. Therefore, the results obtained in this study indicate that the proposed FPE methodology may be a powerful and time-efficient approach for predicting the ensemble average and variance behavior, in both space and time, for the unsteady open-channel flow process under an uncertain roughness coefficient, hence being an approach that would be essential for engineering flow problems."*

**C10:** *The discussion of the results lacks some points or is partly in contrast to what the figures show. Figures 3c,d show a slight decrease of discharge at early times with the FPE method, how do you explain that behaviour? Please explain the offset of the standard deviation at time t=0 for the FPE results. In figure 8c, the FPE result does not reproduce the decrease of the standard deviation. How do you explain the variation of the standard deviation at early times from the MC simulations in figure 8d? Figures 9 and 10 show more of a qualitative match of the results, than a quantitative match.*

**R10:** The referee's questions are addressed by adding some discussion and adjusting some discussion within Section 4. Small changes have been made in several locations, but the main addition is detailed below.

Specific author changes

Discussion has been added in between the two sentences shown on Page 11: Line 7; the added text provides the following explanation:

> *"However, the standard deviation of the FPE methodology shows an offset at time t = 0 when compared to the MC simulations, which can also be noticed at the upstream positions of Fig. 9. Recall that the initial and upstream flow discharge is assumed to be known. Nonetheless, a single known value of the flow discharge, when joined with a spread of roughness coefficients due to the uncertainty involved, leads to an unavoidable spread in the velocity and depth values. Since α and β are functions of the velocity and celerity (and in turn, depth), this spread is translated onto the α–β plane of the FPE methodology. As a result, with an uncertain roughness coefficient, the only way to numerically represent a deterministic discharge in the α–β plane is to have a spread of probability mass over the values involved. The existence of this spread on a non-continuous, discretized α–β plane may have had the most contribution to the offset of the standard deviation at the initial times and positions of Figs. 9 and 10."*

**C11:** *Your outlook on p. 13, l. 16-20 rather belongs to the companion paper. What about faster or more accurate numerical schemes? How could the discrepancies be reduced?*

**R11:** The outlook mentioned by the referee has been removed from this paper and moved to the companion paper, along with additional details elaborating on the expansion of the methodology to problems with different sources of uncertainty, as recommended by Referee #1 in **C9** for the companion paper. Moreover, a new outlook was added to this paper following the suggestion of Referee #1 above; it is detailed below.

Specific author changes

The final paragraph of the conclusion has been changed to read as follows:

> *"    While the FPE methodology satisfactorily described the ensemble average and variability of the open-channel flow system in this study, this methodology is open to improvements especially with regard to reducing any discrepancies in its numerical results. Running a more comprehensive version of the FPE methodology, by including only some of the simplifying assumptions used in this study, may be one option. Another option may involve using a higher-order and more accurate numerical scheme for the discretization of the multidimensional FPE. As such, numerous opportunities present themselves for future research within this topic, all of which would be of great benefit in the further improvement of the proposed methodology."*

**Technical Corrections**

**C12:** *The tense of your abstract makes it read like a summary, please change the tense accordingly.*

**R12:** As per the referee's suggestion, we adjusted the tense of the abstract by changing it from the past to the present tense.

**C13:** *In the abstract you write that the total simulation period of the FPE method is smaller than that of the MC approach. You certainly mean computational time.*

**R13:** It is true that we meant the computational time of the FPE method is smaller than that of the MC approach, and we thank the referee for his/her input regarding this point. We made the necessary adjustment accordingly.

**C14:** *Check the indentations at the beginning of chapters and after equations, please delete them.*

**R14:** Indentations have been deleted from beginning of chapters and after equations.

**C15:** *On p. 2, l. 31 you write that you do not limit the working space of the parameter space. What about parameter combinations, where the neglected cross-covariance terms become large or the system shows a memory?*

**R15:** As may be seen from Kavvas (2003), the uncertainty in multiple parameters (for example, besides the roughness parameter, also an uncertainty in bedslope, etc.) is accounted for within the resulting Lagrangian–Eulerian Fokker–Planck Equation (LEFPE) for the targeted stochastic process in terms of each parameter's variance and in terms of the cross-covariances of the uncertain parameters. Due to the underlying cumulant expansion theory that was used in the derivation of the particular form of the LEFPE for the stochastic Saint-Venant open channel flow, there is no limitation on the size of the variances and cross-covariances of the uncertain parameters. The only limitation for the use of the LEFPE is that the covariance times of the uncertain quantities in the modeling system must be finite.

**C16:** *You write about the conservation of particles, although no particles where introduced in your paper, please reformulate.*

**R16:** We thank the referee for their input regarding this point, since it is true that no specific discussion about particles occurs in this manuscript. Noting that the FPE can be considered as the conservation equation for probability mass, we reformulated the phrase in the manuscript that mentions "particles" and changed it from "to prevent any particles from leaving the domain" to "to prevent any probability mass from leaving the domain".

**C17:** *On p. 2, l. 6 you write that parameters become random through uncertainties. You should write that the parameters are formulated as random functions in order to capture the uncertainties.*

**R17:** The phrasing has been adjusted following the referee's comment.

**C18:** *How did you ensure that Manning's coefficient never fell below 0.01, as described on p.6, l. 27?*

**R18:** We included the possibility of specifying a cutoff value in our code when generating random values of Manning's roughness coefficient ($RN$), and so, ideally a cutoff value of 0.01 can be easily specified to truncate any generated $RN$ values that are under 0.01. However, the mean ($\mu_{RN}$ = 0.035) and standard deviation ($\sigma_{RN}$ = 0.005) that we used for $RN$ in this study allowed us to generate $RN$ values that remained quite far from the 0.01 cutoff value. In fact, from the 68-98-99.7 rule, we find that 99.7% of the values would lie between $RN$ values of 0.02 and 0.05 (i.e., $\mu_{RN} \pm 3\sigma_{RN}$). Therefore, while generating $RN$ values for this study, we never had the need to truncate or discard any of the generated $RN$ values because none of them ever fell below 0.01.

Specific author changes
The sentence of Page 6: Lines 26-28 has been adjusted as follows:

*"Moreover, the selected mean and standard deviation allowed the generation of n values which never fell below 0.01, thus complying with the fact that the roughness coefficients for flows in natural streams and excavated channels are always greater than 0.01 (Chow, 1959)."*

**C19:** *The arguments of a PDF are usually separated by a semi-colon into arguments for which the PDF is a density and normal arguments, e.g. Pope (1985).*

**R19:** This has been corrected in all the respective equations following the referee's comment.

**C20:** *The notation of the m-dimensonal delta function is confusing, drop the exponent m.*

**R20:** The expression of Eq. (9) has been adjusted following the referee's comment.

**C21:** *P. 9, l. 11: What other form of energy, besides kinetic energy is dissipated due to shear stresses?*

**R21:** Besides kinetic energy, the other form of useful fluid energy which is dissipated due to shear stresses is potential energy. It has been added to the manuscript.

**C22:** *P. 9, l. 18: change "very minimal"*

**R22:** The phrase containing "very minimal" was deleted, following the suggestion of Referee #2 in **C11**.

**C23:** *P. 13, l. 9: I do not think that the results for the PDFs are satisfactory in general. In some cases they are, in others they are not.*

**R23:** We understand the referee's point concerning the PDFs. However, we reiterate the novelty of the technique used in this study and the equations derived to tackle the unsteady open-channel flow problem in a purely probabilistic manner, which has not been previously done for this hydrologic process. As such, the PDF results, while not as good as may be desired, may still be considered acceptable results, considering that the method is capable of providing the user with not only the ensemble average, but also the general behavior of the system variability in an efficient amount of time. Also, with such a promising methodology, there is room for improvement especially concerning the numerical method used.

**References**

Chow, V. T.: Open-channel hydraulics, McGraw-Hill civil engineering series, McGraw-Hill, New York, 680 pp., 1959.

Kavvas, M. L.: Nonlinear hydrologic processes: conservation equations for determining their means and probability distributions, J. Hydrol. Eng., 8, 44-53, doi:10.1061/(Asce)1084-0699(2003)8:2(44), 2003.

Sturm, T. W.: Open channel hydraulics, McGraw-Hill series in water resources and environmental engineering, McGraw-Hill, Boston, 493 pp., 2001.

Viessman, W., Knapp, J. W., Lewis, G. L., and Harbaugh, T. E.: Introduction to hydrology, 2nd ed., Series in civil engineering, IEP-Dun-Donnelley, Harper & Row, New York, 704 pp., 1977.

---

## Author Comment (AC2) · 1 Feb 2018

**Response to the comments of Anonymous Referee #2 published on January 5, 2018 concerning the manuscript with reference number: hess-2017-394.**

We would like to thank Referee #2 for his/her insightful feedback. Our responses to the specific points raised by the referee are provided below. Please note that the referee's comments will be presented in italics, preceded by a "**C**", while the corresponding authors' responses will be presented in normal typeface with a blue font, preceded by an "**R**". For some responses, text which was changed or added to the manuscript (based on suggestions from the referee comments) is quoted and placed under "Specific author changes". Please note that the pages and line numbers provided in this document are from the original version of the manuscript.

**C1:** *The authors present a novel way of using the Fokker–Planck Equation (FPE) to derive directly (one simulation) the probability distributions of velocity and depth resulting from uncertain roughness in a hypothetical unsteady open-channel flow problem. Although the efficiency gain over Monte Carlo simulation for the particular case presented seems limited, improving direct ways for probabilistic modelling is a relevant contribution.*

**R1:** We would like to thank the referee for this review.

**C2:** *The paper is well-written and well-structured, with sometimes a bit too many reminders of the story-line and mentioning in an early stage already the main conclusions (e.g. P.2 l.28).*

**R2:** We thank the referee for the positive comment. We would like to note that we worked on adjusting the manuscript to remove any repetitions and obvious statements, including those specified by the referee in the Detailed Comments below.

**Main Comments**

**C3:** *The title is confusing because of the "Ensemble modeling", whereas the main objective of the authors is to present a single simulation solution for providing a pdf. I suggest changing the title of this paper and the companion paper, taking out the term "Ensemble". (e.g. into something like "Fokker-Planck modelling of stochastic open-channel flow in term..", or "Deterministic modelling of..")*

**R3:** We thank the referee for his suggestion and we understand the confusion regarding the title. We will certainly consider the referee's suggestion of improving the title for both papers.

**C4:** *I would suggest to continue reporting and discussing the results for velocity and depth also in the latter part of the Results section (even if only in text, because with figures it would become too long), rather than only discussing discharge results. For velocity and depth, differences are likely to be larger and may lead to better understanding of what are the causes, because in discharge differences in velocity and depth may be cancelled out.*

**R4:** We thank the referee for his suggestion. Following the referee's recommendation in the above comment, as well as in **C14** and **C16** below, we have added a paragraph to the manuscript to briefly

discuss the results of the standard deviation for the velocity and depth. We do not include their figures as to not lengthen the manuscript as mentioned by the referee, but we describe their behavior and provide ranges for their values and some relative differences as suggested by the referee in **C16**.

Specific author changes
A paragraph that addresses the referee's comment is added prior to Page 11: Line 32, and reads as shown below. Note that due to an additional figure, which became denoted as Figure 8 (see Referee #1 **C5**), all figure numbers starting from 8 and above (in the original manuscript) have been increased by one for the revised manuscript. The numbers in the below paragraph refer to the new figure numbers.

" *Concerning the standard deviation of the velocity and depth, it is important to note that their behavior over position is somewhat different from that of the flow discharge. In fact, the standard deviations of the velocity at the same four time positions of Fig. 9 seem to be relatively constant at each time position, having a value between 0.015 and 0.02 $m^3$ $s^{-1}$. On the other hand, the standard deviation of the depth showed a greater range of values at each time position, as a function of location, with values ranging between 0.15 and 0.5 m. When looking at the standard deviations as a function of time at the same four locations of Fig. 10, both standard deviations seem to show that their values increase to reach a maximum and then decrease to levels similar to original levels, not unlike their corresponding ensemble average plots over time. Again, the range of change in the standard deviation of the velocity is much smaller (0.015 to 0.02 $m^3$ $s^{-1}$) than that of the depth (0.15 to 0.5 m). Note that the relative differences of the FPE results when compared to the MC results reach up to 23% and 29% for velocity and depth, respectively.*"

**C5:** *Please include a sensitivity analysis of the MC results with respect to the number of iterations. It would be interesting to check if with more simulations the results go nearer to the FPE result or further away (or no difference), and if with fewer simulations the same result is achieved. This is relevant for the claim on computational efficiency, as also pointed out by Referee#1 (fifth specific comment).*

**R5:** We kindly refer the referee to our reply to **C6** of Referee #1. In our reply, we present a sensitivity analysis showing how the number of realizations of the MC simulations affects the results of the ensemble average and standard deviation of the flow discharge, thus explaining in greater detail the choice of 1000 simulations in this study. We also mention that we believe that including such a long discussion and sensitivity analysis in the manuscript may cause a digression from the main idea of the manuscript, which is mainly to gauge the performance of the FPE methodology. Therefore, we believe that it would be preferable not to add such an analysis to the manuscript. Nonetheless, we include some text in the manuscript to clarify our choice of 1000 simulations to the reader, and we briefly mention the lower accuracy occurring at lower MC realizations (please see "Specific author changes" in our response to **C6** of Referee #1).

**C6:** *The analysis and discussion on computational time needs to be more detailed (including computational times, hardware used, etc.) and expanded. In particular with whether the FPE approach is suitable for parallelisation, if not, the MC analysis, for the case study presented, can be easily made more efficient. The authors could perhaps also include their expectations on the applicability and computational efficiency of their FPE method for larger systems. Would the gain with respect to MC increase or not?*

**R6:** More details have been given in the revised manuscript for the discussion of computational time needs as suggested by the referee. Moreover, a short discussion regarding the parallelization of the FPE as well as the greater advantage of the computational efficiency of the FPE methodology for systems with greater numbers of uncertainties is also added to the manuscript. These additions will be detailed below. The referee is also kindly referred to our replies to Referee #1, especially regarding **C6** (last two paragraphs) and **C9.**

Specific author changes
Technical information has been added directly before Section 3.1 (on Page 6: Line 14):

*"Note that all simulations for the MC approach and the FPE methodology were run on a computer having 16 GB of RAM and an Intel i7 processor with four cores, each core having a base frequency of 2.40 GHz and a maximum frequency of 3.40 GHz."*

Two paragraphs were added to the end of the Discussion section, and read as follows:

*"    Moreover, it should be noted that these FPE results required a significantly less amount of time for computation as opposed to the MC results. Recall that the 1000 MC simulations were parallelized and run over all four cores (with no hyperthreading), thus noticeably reducing the computational time as compared to an un-parallelized run. With such parallelization, the MC simulations ran for over 2 days. On the other hand, the results of the FPE methodology, which was not parallelized, were obtained in about 7 hours.*

*    If we observe the computational times of the implicit numerical solution of the FPE methodology, the portion of the simulation requiring the greatest time is filling out the coefficient matrix, especially for small α and β discretizations. Parallelizing this portion over the four cores would allow one to considerably reduce the time to fill out the coefficient matrix, thus reducing the total computational time of this method. Without the parallelization of the FPE methodology, its one simulation may still not seem to provide an immense advantage when only one uncertain parameter is involved, especially with the possibility of parallelizing the MC simulations among a much larger number of cores. Nonetheless, when the problem being solved involves a greater number of uncertain parameters and boundary conditions, or even a larger system, such an advantage may prove to be crucial. In fact, the computational expense of the MC simulations for such a case would be expected to increase exponentially due to the higher number of simulations needed to maintain the desired accuracy in the results, thus significantly increasing the computational time regardless of parallelization. On the other hand, such additional uncertainties can be easily implemented into the FPE methodology by making simple changes and additions that will be reflected in Eq. (5), after which the FPE would be solved following similar steps as discussed for this study, with minimal implications on the computational expense."*

**C7:** *The gain in computational efficiency, as presently described, seems limited. Hence, the claimed contribution there, in Abstract and Conclusions, should be down-sized or contextualised.*

**R7:** The referee's point is acknowledged, and adjustments have been made to the corresponding text in the Abstract and the Conclusion as detailed below.

Specific author changes
In the Abstract, Page 1: Line 16-17 was adjusted as follows:

*"Moreover, the total computational time of the FPE methodology is smaller than that of the MC approach, which could prove to be a particularly crucial advantage in systems with a large number of uncertain parameters."*

In the Conclusion, Page 13: Lines 12-13 now read as follows:

*"Moreover, the FPE methodology results were obtained by running only one simulation, as opposed to the large number of simulations performed by the MC approach. Such an advantage becomes prominent with a greater number of uncertain parameters and boundary conditions, in which case the computational expense of the MC simulations that is needed to preserve the desired accuracy would exponentially increase. On the other hand, only simple adjustments would be required for the FPE, which could then be solved as was done in this study, with minor implications on its computational expense."*

**Detailed Comments**

**C8:** *P.2 l.28: "..producing the complete ensemble model results.." is not correct, because, if I understand correctly, the method does not reproduce the individual traces (ensemble members). Therefore, this should be something like "..producing the statistical properties.."*

**R8:** The sentence has been adjusted following the referee's comment.

**C9:** *P.7 l.14-15: Explain the choice of 1000 simulations. Report the sensitivity of the statistical characteristics to the number of simulations in the MC.*

**R9:** We kindly refer the referee to our reply to **C6** of Referee #1 which expands on this topic and explains in greater detail the choice of 1000 simulations.

**C10:** *P.8 l.27-28: Repetition. There is already a sentence connecting Sections 3 and 4 in lines 23-25. Consider leaving out one of the two.*

**R10:** Following the referee's comment, the sentence at the end of Section 3 has been removed to reduce the repetition.

**C11:** *P.9 l.18: Repetition. Delete ".., with very minimal differences among the two"*

**R11:** As suggested by the referee, the phrase "with very minimal difference among the two" was deleted.

**C12:** *P.9 l.32-33: However, ... ,but... Consider reformulating.*

**R12:** As suggested by the referee, the sentence was reformulated as detailed below.

Specific author changes
The reformulated sentence now reads as follows:

*"Similarly to Fig. 4, a slight overestimation can be noticed from the FPE methodology especially around the peak depths, but with a maximum relative difference of only around 7.5%."*

**C13:** *P.10 l.13-17: Reformulate removing redundancies. (Or consider leaving out, because it reads perhaps too much as general conclusions, while this is in the middle of presenting and discussing results)*

**R13:** The sentences noted by the referee have been left out.

**C14:** *P.10 l.20: Why do the authors continue only with Discharge? Differences in velocity and depth may be cancelling each other in the resulting discharge. Also when thinking of flood risk management applications, it may be more interesting to look at velocity and depth variance.*

**R14:** As suggested previously by the referee, we provide additional text in the manuscript to briefly discuss the results for the standard deviations of the velocity and depth. We kindly refer the referee to our reply to **C4** for our full response.

**C15:** *P.10 l.18-23: Too much repetition. Suggest to shorten and merge with next paragraph where actually the presentation of variability results starts.*

**R15:** As suggested by the referee, the noted paragraph was shortened to one and merged with the following paragraph as follows:

*"In a similar manner to the ensemble averages, the relative performance of the FPE methodology in predicting the system's variability was examined, this time by checking the standard deviations."*

**C16:** *P.10 l.33: "relatively small" Suggest to add some of the differences in %. Also provide differences in standard deviation for velocity and depth. P.11 l.15-21: The results for velocity and depth may help in understanding the causes of differences in variability.*

**R16:** The phrase structure for the sentence of Page 10: Line 33 has been changed due to changes from other comments. The percent relative differences have been added for the standard deviation for velocity and depth as suggested by the referee. Again, we kindly refer the referee to our reply to **C4** for our full response.

**C17:** *P.11 l.27-31: As described in main comments above, please expand the analysis and discussion of computational efficiency, and make it a separate paragraph.*

**R17:** We kindly refer the referee to our reply to **C6** above for the full response regarding this matter.

**C18:** *P.12 l.18: General sentence. Consider deleting.*

**R18:** As suggested by the referee, the sentence has been deleted.

---

## Author Comment (AC3) · 1 Feb 2018

**Response to the comments of Dr. Alberto Bellin from the Short Comment published on September 12, 2017 concerning the manuscript with reference number: hess-2017-394.**

We would like to thank Dr. Bellin for his insightful comments. Our responses to the specific points raised by him are provided below. Please note that his comments will be presented in italics, preceded by a "**C**", while our responses will be presented in normal typeface with a blue font, preceded by an "**R**". For some responses, text which was changed or added to the manuscript (based on suggestions from the comments) is quoted and placed under "Specific author changes". Please note that the pages and line numbers provided in this document are from the original version of the manuscript.

**C1:** *In this manuscript the authors present an illustrative example of the Fokker Planck equation associated to the De Saint Venant equation for a spatially uniform but random roughness coefficient.*

**R1:** We thank Dr. Bellin for his review of this companion paper.

**C2:** *As I already commented in my review of the previous manuscript I think that the material is not enough to justify a two-paper series. Therefore, my suggestion is for merging the two manuscripts into one. The first part (up to page 5) of the second manuscript summaries what already presented in the first one and is not longer needed in the merged manuscript, while the second part can be easily merged with the first manuscript.*

**R2:** We again thank Dr. Bellin for his comment and concern regarding this matter. However, as we have mentioned in our reply to comment **C2** from his review of our companion paper (Referee #2 for hess-2017-393), we still believe in the ability for both manuscripts to remain as two standalone, companion papers. We would also like to note that neither Referee #1 of the companion paper, nor Referees #1 and #2 of this manuscript have presented any concerns regarding having two standalone manuscripts, nor have they presented the desire to join the two manuscripts into one. Below is our original reply to Dr. Bellin's comment **C2** from our companion paper regarding the same matter:

> *"We thank the referee for the comment. The ensemble-averaging technique used in this study was indeed developed by Kavvas (2003) and this technique has been applied to other processes with different governing equations where Fokker–Planck Equations (FPEs) specific to those processes were obtained and applied successfully. However, this technique had never been applied to the Saint-Venant equations to tackle the stochastic unsteady open-channel flow problem. As such, the novelty of the proposed FPE methodology that was developed in this manuscript was, firstly, to figure out how to apply the Kavvas (2003) technique to the Saint-Venant equations especially through the transformations that provided us with the state variables α and β and that allowed us to write the Saint-Venant equations as four Ordinary Difference Equations (ODEs) in the specific forms of Eqs. (15) to (18) (for this was not a straightforward process), and secondly, to go forth with developing the FPE that is specifically for the stochastic unsteady open-channel flow process, an equation which has not been developed before. Hence, this study clearly derives and presents an entirely new FPE that can be used to solve for the probability density of the state variables of a stochastic open-channel system, which is not found elsewhere in the literature. And while the numerical discretization was made following the Chang and Cooper (1970) scheme, the scheme was generalized from its original one-dimensional form and adapted to the four-dimensional FPE that was being solved in this study. Therefore, joining this manuscript and the companion manuscript*

*into one manuscript, while placing a large portion of this first manuscript in the appendix, would mostly take away from the importance of these equations and from the work that was done in arriving to those equations. As such, we believe in the novelty of the equations derived in this manuscript, and thus we believe in its ability to stand as its own manuscript. We would also like to note that Referee #1, who has read and reviewed both manuscripts, has not mentioned any desire for the joining of the two papers into one, in which case we would assume that Referee #1 may not have seen any concern with them being two standalone, companion papers."*

**C3:** *The boundary condition used at the lower end of the channel in the solution of the de Saint Venant Equation in a Monte Carlo framework are unclear and should be better specified and justified. At page 7 line 12 I read "... while at the downstream end, the channel was assumed to be hydraulically long so that the flow can be taken as normal flow, thus satisfying the Manning's equation". This sentence is unclear: what is normal flow? Mathematically the downstream condition can be of imposed water depth y, with the velocity V obtained through the Manning's equation, or imposed velocity (or water discharge) and again the Manning's equation provides the water depth. The imposed condition can be either stationary, i.e. constant, or transient. It seems from the results that the authors choose the second option (Figures 2, 3 4 and 5 shows that the ensemble mean of both y and Q changes with time at the lower end of the channel), but no details on the specific boundary conditions are provided in the text. Assuming that one of the above two boundary conditions have been selected at the lower end of the channel, this choice should reflect to the boundary condition for the stochastic variables in the FPE equation. For instance, if y is imposed its PDF is a Dirac delta, while the pdf of the velocity is related to the pdf of the roughness coefficient through the Manning's equation. In turn, the PDFs of α and β depends from the PDFs of y and V through equations (13) and (14) of the first manuscript. Similar arguments can be applied if the BC is of imposed water discharge (i.e. velocity) and the water depth is computed through the Manning's equation. What puzzles me is that the authors impose a reflection boundary at the end of the channel (line 15, page 8), which is apparently not compatible with the previous conditions and with those imposed in the MC simulations. Since Figures from 2 to 5 show clearly that the water wave interacted with the downstream boundary, I am expecting that the boundary condition here should have an impact on the solution. This reinforce the need to select compatible boundary conditions in both models.*

**R3:** We thank Dr. Bellin for this comprehensive and crucial comment regarding the downstream boundary condition (BC). We certainly agree with his final sentiment regarding the importance of selecting a compatible downstream BC for both the MC and FPE models. Regarding the first point Dr. Bellin raises, it is true that the downstream BC may be imposed as water depth ($y$) or as velocity ($V$). However, to add to this, we provide a quote from the book *Open Channel Hydraulics* by Sturm (2001) specifying that "the downstream boundary condition in a flood routing problem also might be a stage or discharge hydrograph, but in some cases, it could be a depth-discharge relationship." Hence, concerning the last part of Page 7: Line 12 which Dr. Bellin indicated, what we meant by normal flow was that the Manning's equation is assumed valid at this downstream boundary, and that our downstream BC is not a specified/imposed depth or velocity, but instead it is the depth-discharge relationship represented by the Manning's equation. The way we computed the depth and velocity downstream will be explained in the following paragraph.

Recall that the manner with which we solve our MC simulation is through the method of characteristics, which is explained in detail in Viessman et al. (1977) and Sturm (2001). We describe here, in short, the steps used in order to compute the values of the flow variables at the downstream boundary. When using the method of characteristics, the solution is usually found at the intersection of the forward ($C_1$)

and backward ($C_2$) characteristic curves. The point of intersection is denoted by $P$. However, when calculating the values for the downstream boundary condition, only $C_1$ is used (Eq. (1)), along with its compatibility equation (Eq. (2)). Now, for the downstream BC, the unknown value is at the intersection of $C_1$ with the downstream boundary (again this point is denoted by $P$). $C_1$ originates from a point from the previous time step, and this point (denoted by $L$) is upstream of point $P$. We use the known flow variables at point $L$ ($V_L$ and $y_L$) from the previous known time step ($t_L$) located at a known location ($x_L$) in order to compute $t_P$ from the equation of $C_1$ (Eq. (1)). Then, from the compatibility equation along $C_1$ (Eq. (2)), we get a nonlinear expression in depth ($y_P$) which results from the substitution of $V_P$ with its expression from Manning's equation as a function of $y_P$. As such, this nonlinear expression is solved to find $y_P$ which is then used to compute $V_P$. Therefore, while no imposed values for $y_P$ and $V_P$ are set at the downstream boundary, the expression of the Manning's equation in combination with the equations corresponding to the forward characteristic curve $C_1$ are used to determine these variables.

Since none of the flow variables ($y$ or $V$) are imposed at the downstream boundary conditions, we do not need to specify any Dirac delta function for these variables at the downstream boundary. Moreover, we would like to mention that the reflecting boundary conditions for the FPE method were used to describe the boundaries of the $\alpha$ and $\beta$ dimensions, as mentioned in the original manuscript on Page 8: Lines 17-19. However, for the upstream boundary condition, the probability densities were known. Finally, the downstream boundary condition for FPE was formulated in a way that would replicate the downstream BC of the MC model with its depth-discharge relationship (Manning's). Moreover, in the manuscript we note that we also extend the downstream boundary condition much further than the required reach length in the downstream direction in order to eliminate any of its effects on the numerical solution.

Specific author changes
The manuscript text describing the downstream boundary conditions for the MC and FPE simulations have been adjusted, and the new versions are shown below.

The text of Page 7: Lines 11-13 (for MC) now reads as follows:

*"As for the boundary conditions, the flow hydrograph at the channel entrance was given, while at the downstream end, the channel was assumed to be hydraulically long so that the flow can be taken as normal flow, thus satisfying Manning's equation. As such, the downstream boundary condition was chosen as the depth-discharge relationship represented by Manning's equation. This equation, along with the equations corresponding to the positive characteristic (i.e., Eqs. (1) and (2)), was used to compute the flow variables at the downstream boundary."*

The text of Page 8: Lines 20-22 (for FPE) now reads as follows:

*"As for the upstream boundary, recall that the discharge hydrograph was assumed to be known upstream, in which case the probability densities at the upstream boundary of the x–dimension were known for all t > 0. Finally, the downstream boundary in the x–dimension was formulated to replicate that of the MC model, and it was extended downstream much further than the required length of the reach so as to eliminate any of its effect on the numerical solution."*

**C4:** *In addition, given the boundary condition of the FPE at the initial section I was expecting here the standard deviation of the water discharge equal to zero, as in the MC simulations. However, this is not the case, as shown in Figures 8 and 9. This unexpected result needs justification.*

**R4:** We kindly refer Dr. Bellin to our reply to **C10** of Referee #1 of this manuscript, which includes our full response regarding the same matter.

**References**

Sturm, T. W.: Open channel hydraulics, McGraw-Hill series in water resources and environmental engineering, McGraw-Hill, Boston, 493 pp., 2001.

Viessman, W., Knapp, J. W., Lewis, G. L., and Harbaugh, T. E.: Introduction to hydrology, 2nd ed., Series in civil engineering, IEP-Dun-Donnelley, Harper & Row, New York, 704 pp., 1977.